# Epidermal electronic-tattoo for plant immune response monitoring

Tianyiyi He [1,2,3,4], Jinge Wang [3,5], Donghui Hu [3,5], Yanqin Yang[1], Eunyoung Chae [3,5,6] ✉ & Chengkuo Lee [1,2,3,7,8] ✉

Real-time monitoring of plant immune responses is crucial for understanding plant immunity and mitigating economic losses from pathogen and pest attacks. However, current methods relying on molecular-level assessment are destructive and time-consuming. Here, we report an ultrathin, substrate-free, and highly conductive electronic tattoo (e-tattoo) designed for plants, enabling immune response monitoring through non-invasive electrical impedance spectroscopy (EIS). The e-tattoo's biocompatibility, high conductivity, and sub-100 nm thickness allow it to conform to leaf tissue morphology and provide robust impedance data. We demonstrate continuous EIS analysis of live transgenic *Arabidopsis thaliana* plants for over 24 h, capturing the onset of NLR-mediated acute immune responses within three hours post-induction, prior to visible symptoms. RNA-seq and tissue ion leakage tests validate that EIS data accurately represent the physiological and molecular changes associated with immune activation. This non-invasive tissue-assessment technology has the potential to enhance our comprehension of immune activation mechanisms in plants and paves the way for real-time monitoring for plant health management.

Plant diseases pose a threat to global agriculture, with the Food and Agriculture Organization (FAO) estimating annual economic losses of approximately $220 billion[1,2]. Plant immunity is the major coping strategy that plants make use of to counteract various diseases and attacks[3]. The ability to monitor and assess plant immune responses in real-time is critical for enabling timely and precise interventions, as well as for elucidating the fundamental mechanisms underlying plant immunity[4–6]. However, achieving real-time monitoring remains a pressing and unresolved challenge within the fields of agriculture and plant science. Although advanced molecular and genetic techniques such as transcriptomics, proteomics, and metabolomics have extensively explored the underlying mechanisms and signaling pathways,

these approaches often involve destructive, non-dynamic, and time-consuming sampling and analysis methods[7,8]. Such limitations hinder the capture of rapid and nuanced changes occurring during immune responses. For instance, the electrolyte leakage assay, a common technique in plant physiology to evaluate cell membrane integrity and damage, involves mechanical punching to obtain leaf disks and immersing them in ultrapure water for electrical conductivity measurement[9]. This method not only damages the leaf samples but also fails to capture time-evolving variations.

Flexible sensors have enabled high-fidelity measurements on complexly shaped and deforming surfaces, which excel in various applications[10,11], such as sports monitoring[12,13], healthcare

[1]Department of Electrical and Computer Engineering, National University of Singapore, Singapore 117583, Singapore. [2]Center for Intelligent Sensors and MEMS (CISM), National University of Singapore, Singapore 117608, Singapore. [3]Research Center for Sustainable Urban Farming (SUrF), National University of Singapore, Singapore 117558, Singapore. [4]Artificial Intelligence Research Institute, Shenzhen MSU-BIT University, Shenzhen 518172, China. [5]Department of Biological Sciences, National University of Singapore, Singapore 117558, Singapore. [6]Department of Biology, University of Oxford, South Parks Road, Oxford OX1 3RB, UK. [7]NUS Graduate School - Integrative Sciences and Engineering Programme (ISEP), National University of Singapore, Singapore 119077, Singapore. [8]National University of Singapore Suzhou Research Institute (NUSRI), Suzhou Industrial Park, Suzhou 215123, China. ✉e-mail: dbsce@nus.edu.sg; elelc@nus.edu.sg

monitoring[14,15], rehabilitation[16,17], and robotics[18,19]. Recently, flexible sensors tailored for plants have emerged, offering non-invasive methods to monitor various aspects of plant physiology by detecting strain, temperature, humidity, volatile organic compounds (VOCs), light, and electrical impedance spectroscopy (EIS)[20–30]. Among these, EIS stands out for its unique capability to assess biological tissues noninvasively by applying alternating electrical fields. For example, Jae et al. demonstrated the use of vapor-printed polymer electrodes on living plants (pothos and hosta) and detached leaves of fruiting plants for EIS analysis, successfully showing its feasibility for detecting leaf tissue damage[26,28]. However, the process of depositing these electrodes, which involves exposure to a 100-mtorr vacuum in a reactor for 20 min, poses significant challenges for practical applications. This method also has limitations for delicate plant species or large pots, restricting its broader use in the extensively studied model plant species such as *Arabidopsis thaliana (A. thaliana)*. To achieve non-invasive and efficient EIS analysis across a wide range of plant species, it is crucial to address several key challenges, including improving electrode material conductivity, ensuring conformity to leaves with diverse dermal structures, and minimizing interference with plant health and physiological processes.

In this work, we report an ultrathin, substrate-free, and highly conductive electronic tattoo (e-tattoo) designed for plants, capable of real-time and non-invasive measurement of plant immune responses with robust EIS analysis (Fig. 1a-d). The plant e-tattoo consists of substrate-free silver nanowire (AgNW) network, fabricated by a vacuum filtration process and transferred to the leaf surfaces through a gentle and non-invasive in-water transfer printing process. It minimizes any potential harm that could be imposed to the plant during the printing process. The e-tattoo exhibits biocompatibility, high electrical conductivity, and ultrathin structure, which can adapt to the microscopic morphology of leaf tissues via just van der Waals interactions. Consequently, the e-tattoo provides high-quality impedance spectrum data for diverse species, surpassing conventional metal films that have been widely employed for plant EIS analysis. We demonstrate more than 24 h of continuous monitoring of the physiological status of intact *A. thaliana* plants, enabling the successful recording of the hypersensitive response triggered by plant immune receptors within three hours post-activation. RNA-seq analysis and tissue ion leakage tests confirm that the variations in EIS data accurately reflect the physiological and molecular activities involved in the genetically triggered autoimmune response of *A. thaliana*. The technology described herein has the potential to provide deeper insights into the mechanisms governing plant immunity, thereby fostering the development of sustainable agricultural practices in the future.

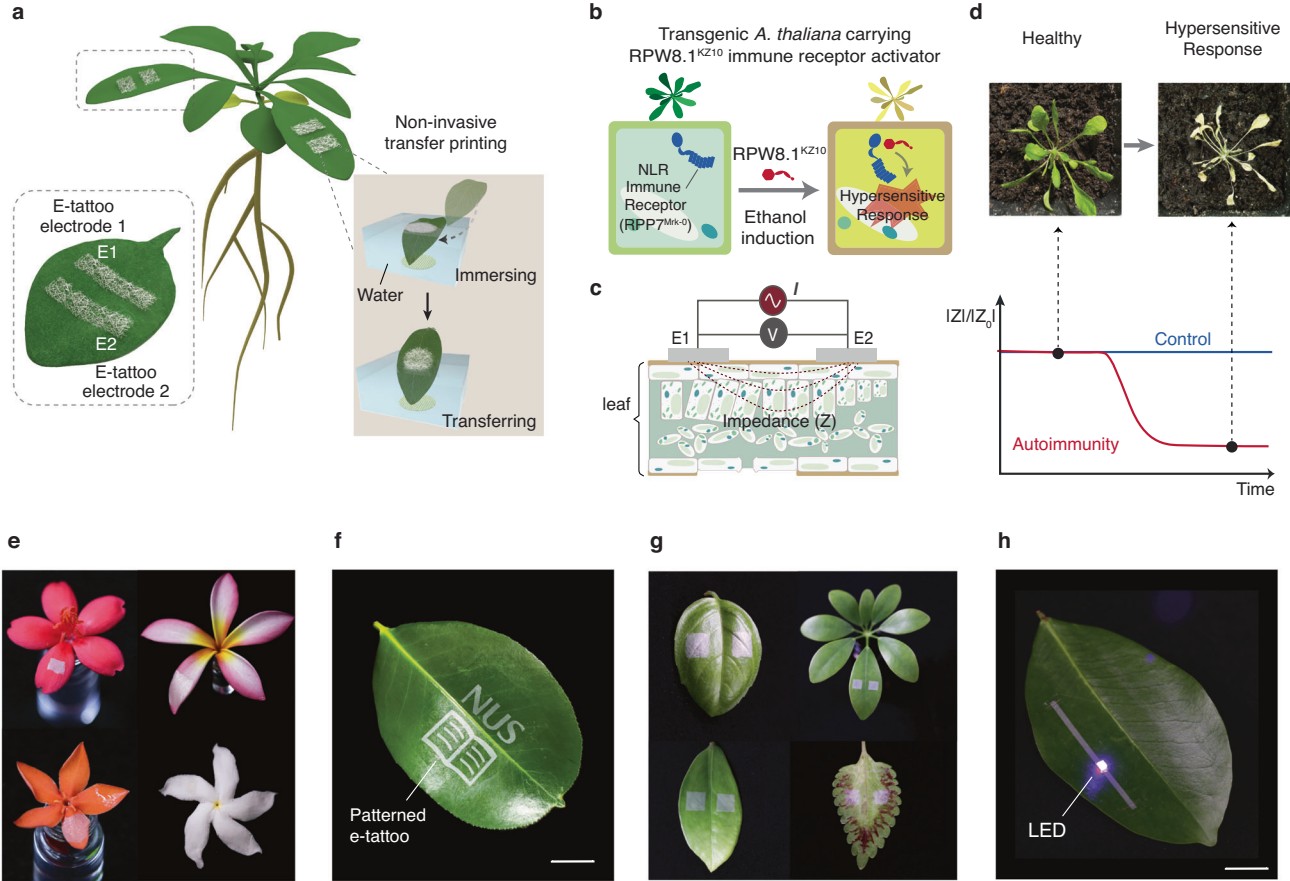

**Fig. 1 | Ultrathin, substrate-free, and highly conductive e-tattoo for continuous immune response monitoring. a** Schematic illustration of the e-tattoo on an *A. thaliana* leaf and the non-invasive in-water transfer printing process. **b** Simplified illustration of *A. thaliana* signaling events in response to the induction of autoimmunity, highlighting the hypersensitive response as a key feature of plant immune responses. The transgenic line in the Mrk-0 accession carries the RPP7 immune receptor that can be activated upon the induced expression of RPW8.1 from KZ10 accession. **c** Schematics showing the operating principle of EIS measurement using the e-tattoo. **d** Relative change of impedance magnitude over time during the autoimmune response. Inset photos present a transgenic *A. thaliana* plant before induction and 7 days post-induction. **e** Photographs of e-tattoos transferred to different flower pedals with no observable harm (top left to bottom right: peregrina, frangipani, ixora, and pinwheel flower). **f** Patterned e-tattoo on a leaf. Scale bar, 1.5 cm. **g** The e-tattoos transferred to different leaves with excellent conformability (top left to bottom right: basil, *Schefflera heptaphylla*, zamioculcas, and *coleus scutellarioides*). **h** An LED lit up through the conductive traces of the e-tattoo, demonstrating the e-tattoo's excellent electrical conductivity. Scale bar, 1 cm.

## Results

### Design of the e-tattoo

AgNWs are emerging as highly promising materials for flexible electronics, including e-tattoos, due to their exceptional mechanical flexibility, electrical and thermal conductivity, and optical transparency[31]. However, achieving a uniform distribution of AgNWs on a substrate poses significant challenges. Variability in film thickness or nanowire density can adversely affect both electrical and optical properties. Additionally, traditional elastomer substrates, such as polydimethylsiloxane (PDMS), though widely used in flexible electronics, are not suitable for plant e-tattoos designed for EIS. These substrates inhibit direct contact between the electrode material and biological tissue and increase the overall thickness of the e-tattoo, thereby compromising its conformability.

To address these issues, we developed a fabrication process to realize e-tattoo consisting of an ultra-thin, uniform, and standalone AgNW film. Our method employs a vacuum filtration process to create the film, which can be released into water, where it floats on the surface, and is subsequently transferred directly onto biological tissues using an in-water transfer printing technique (Supplementary Fig. 1 and Supplementary Movie 1). This approach allows for precise control over the dispersion of AgNWs, resulting in a highly uniform film with an approximate thickness of 100 nm (Supplementary Fig. 2). The film exhibits good optical transparency in the visible light range (400–700 nm), ensuring minimal interference with photosynthetic activity when applied to the adaxial surface of plant leaves (Supplementary Fig. 3). The in-water transfer printing process is sufficiently gentle to be used with delicate tissues such as petals (Fig. 1e). Patterns can be accurately applied to leaf surfaces using a designed shadow mask (Fig. 1f and Supplementary Fig. 4). The film can be configured into a common and simplified two-electrode arrangement, making it suitable for EIS analysis (Fig. 1g). The superior mechanical properties of the AgNW-based plant e-tattoo ensure its effective lamination onto leaf surfaces, even accommodating the presence of trichomes that typically disrupt stable electrode-tissue interfaces. The film exhibits excellent conductivity (Fig. 1h), with a sheet resistance below 5 Ω/square, thereby facilitating high-quality EIS data collection.

### Biocompatibility of the e-tattoo

Ensuring the biocompatibility of plant wearable sensors is paramount when considering their integration with plant tissues. Numerous plant species, including the model species studied extensively in plant science, exhibit sensitivity to mechanical and chemical stimuli. To effectively monitor plant physiological status over extended periods, it is imperative that any sensor attached to biological tissues shall be safe with minimal adverse effects on plant health and functionality crucial for growth and development. While existing plant wearable sensors primarily target robust houseplants (e.g., pothos, *hosta*, *P. macrocarpa*) and large crop plants (e.g., grapevine, watermelon, tomato)[20,22–26,28], investigations into model plants essential for advancing plant biology, such as *A. thaliana*, have been limited. *A. thaliana* is a small dicotyledonous species and a member of the *Brassicaceae* or mustard family, which has been the focus of intense genetic, biochemical and physiological study for over 40 years.

To assess the e-tattoo's biocompatibility, a 7-day attachment study was conducted on intact leaves of *A. thaliana* potted on soil. In this experiment, the test group had an e-tattoo printed on the 7th leaf, while the control group had no e-tattoo. Both groups exhibited ordinary growth and development over the week, with the e-tattoo printed leaf remaining healthy and intact (Fig. 2a and Supplementary Fig. 5). There was no observable difference between the test and control plants, indicating a minimal adverse effect of the e-tattoo and its transfer printing process on the *A. thaliana*. The yellowing of older leaves in Fig. 2a reflects natural leaf senescence during the flowering stage, as nutrients are redirected toward reproductive organs, a process observed in both plants with and without the e-tattoo. To further demonstrate its biocompatibility to a broader range of species, an extended 60-day test was performed on a cut pothos stem, with one leaf having the e-tattoo printed. Over the two-month period, adventitious roots with side shoots and foliage growth were observed, showcasing the e-tattoo's excellent biocompatibility across diverse plant species without observable side effects (Fig. 2b).

The optical microscopy (OM) image of the e-tattoo printed on the pothos leaf demonstrates excellent conformability of the substrate-free e-tattoo (Fig. 2c). Both pristine and e-tattoo-covered leaf surfaces exhibit the same microstructures, including clearly visible stomatal structures, compared to those covered with conventional sensor substrate materials (e.g., a PDMS thin film, as shown in Supplementary Fig. 6). The scanning electron microscopy (SEM) image further highlights the preservation of stomatal functionality, in which nanowires cover stomatal surfaces but not blocking the stomatal pore (Fig. 2d). This feature is beneficial for long-term attachment and monitoring, as stomata are crucial for gas exchange and plant survival. Another critical aspect of biocompatibility is the impact on photosynthesis. Unlike humans, plants rely on light absorption on the leaves for energy production. Thus, the e-tattoo must be thin and not impede light absorption in the visible range (Supplementary Fig. 7). To visualize the effect of potential light blocking on the leaf's health, a two-week attachment test was conducted (Fig. 2e). The tissue under the black tape turned light yellow after two weeks, implying impaired health and reduced chlorophyll content (Supplementary Fig. 8). In contrast, the tissue under the e-tattoo remained unchanged. Fluorescence imaging further verified these variations, showing that the e-tattoo-covered area maintained the same intensity as the rest of the leaf, while the area covered by the black tape exhibited a noticeable increase in intensity (Fig. 2f and Supplementary Fig. 9, 10).

Overall, these findings demonstrate the e-tattoo's excellent biocompatibility with plant tissues, ensuring its suitability for long-term attachment and monitoring without impeding essential physiological processes such as gas exchange and photosynthesis (Supplementary Table 1). These favorable characteristics enable the extended EIS analysis of intact and fragile *A. thaliana* plants, as detailed in the following section.

### Characterization of the e-tattoo

A well-recorded impedance spectrum of leaves requires both conformal attachment and good conductivity of the electrode material. We conducted a comprehensive characterization of the electrical properties of the e-tattoo from various perspectives. Sheet resistance measurements were performed on e-tattoos transferred onto both smooth silicon wafer and diversified leaves, demonstrating that the transferred AgNW film maintains excellent electrical conductivity on surfaces with varying roughness and microstructures (Fig. 3a). The uniformly distributed network of nanowires resulting from the vacuum filtration process leads to comparable sheet resistances across different substrates, highlighting the e-tattoo's versatility for a wide range of species and applications. To assess the flexibility of the e-tattoo, it was transferred to a PET film, and the resistance of the e-tattoo electrode under different bending degrees was recorded. Negligible relative resistance changes were observed when the AgNW-PET film was bent upward and downward from 30 to 180 degrees (Fig. 3b). Dynamic bending tests further confirmed the e-tattoo's resilience, exhibiting negligible relative resistance changes under varied bending conditions (Fig. 3c). When immersed in water, the e-tattoo exhibited minute resistance variation (<0.05%), demonstrating excellent moisture stability (Fig. 3d). This feature is crucial for stable recordings during on-plant tests, as leaves may get wet in actual situations. As the temperature changed from 20 °C to 30 °C, a small relative resistance change was recorded (<1.5%), indicating good stability under typical environmental temperature variations (Fig. 3e).

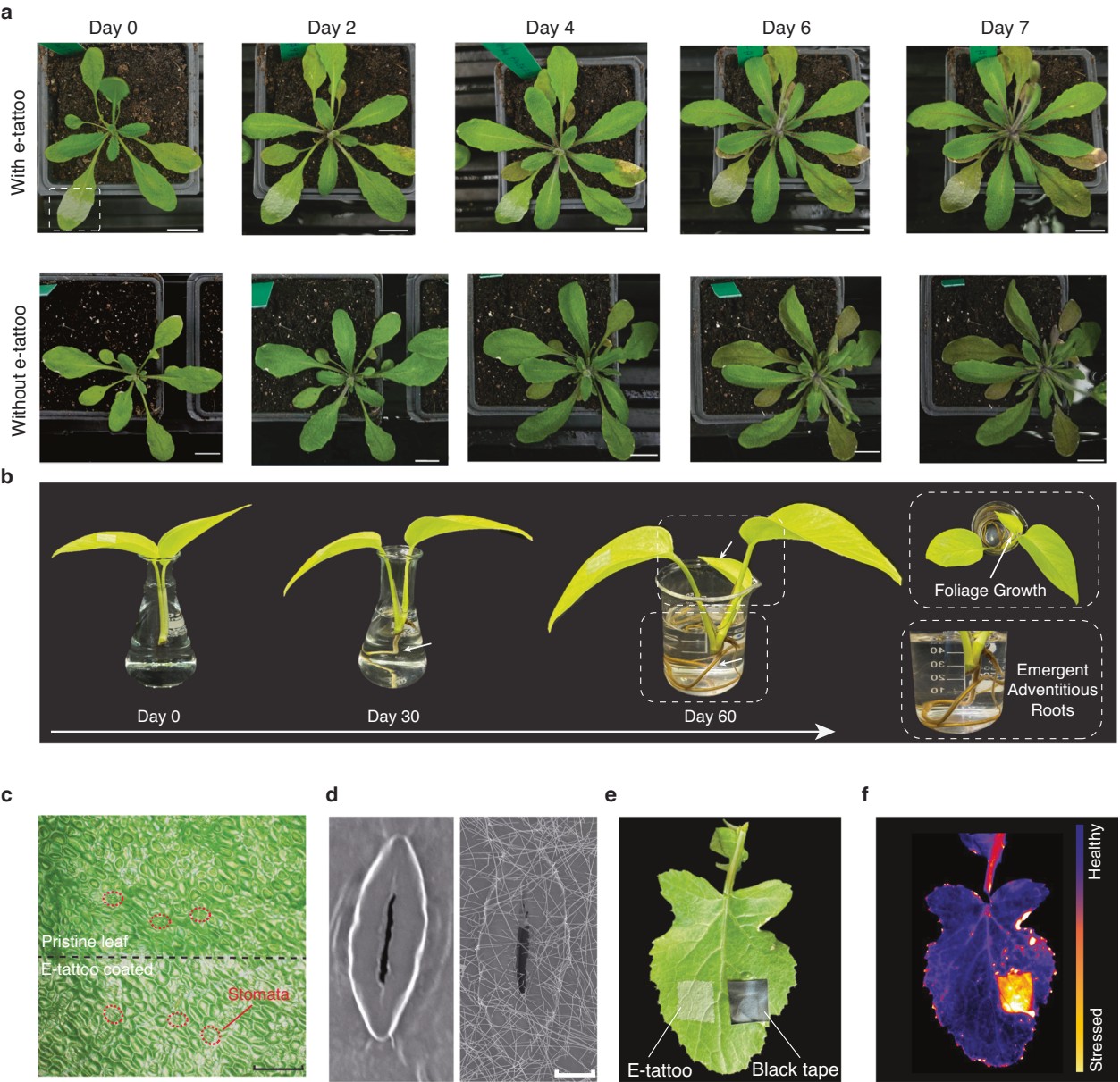

**Fig. 2 | Biocompatibility test of the e-tattoo on different plant species. a** Photos of the *A. thaliana* taken over one week, showing the test group with e-tattoo printed on the 7th leaf and the control group without e-tattoo. Scale bar, 1 cm. **b** Photos of a pothos stem growing in water for two months, with one leaf printed with e-tattoo, exhibiting a new leaf and various adventitious roots after 60 days. **c** Optical microscope image of the abaxial surface of a pothos leaf, half covered by the e-tattoo. Scale bar, 200 µm. Similar results were observed in more than three independent experiments. **d** SEM image of open stomata on the abaxial surface of a pothos leaf. The left panel shows a pristine leaf, while the right panel displays a leaf printed with the e-tattoo. Scale bar, 5 µm. Similar results were observed in two independent experiments. **e** Photo of a *Brassica rapa* (oilseed sarson) leaf with a square e-tattoo printed on the left side and black tape attached on the right side. **f** Fluorescence image of the *Brassica rapa* leaf after removing the e-tattoo and black tape, showing the e-tattoo-covered area (left) and the black tape-covered area (right).

Additionally, long-term stability assessments demonstrated consistent resistance values over a month under ambient conditions, affirming the e-tattoo's durability and reliability (Fig. 3f).

To ensure precise tracking of leaf impedance spectra over extended periods, especially during physiological changes, the e-tattoo must adhere conformably to tissue surfaces and maintain stable electrical properties, even as they deform and wrinkle. The vacuum-filtrated AgNW films, characterized by uniformly distributed nanowires, offer exceptional conformability, conductivity, and mechanical stability. As shown in Fig. 3g, the e-tattoo deforms along with the leaf as it progressively dehydrates and wilts, maintaining conformal attachment even when the leaf is fully desiccated and significantly wrinkled. Throughout this process, minimal fluctuations in relative resistance (<5%) were recorded, highlighting the e-tattoo's capacity to preserve impedance spectra fidelity despite pronounced leaf deformations, thereby ensuring accurate monitoring of dynamic leaf tissue physiological status. To further assess the robustness of the plant e-tattoo in real-world plant environments, we monitored its resistance changes on actual leaves over a three-week period (Fig. 3h). The results revealed minimal variations throughout this timeframe, further validating the e-tattoo's stability and reliability.

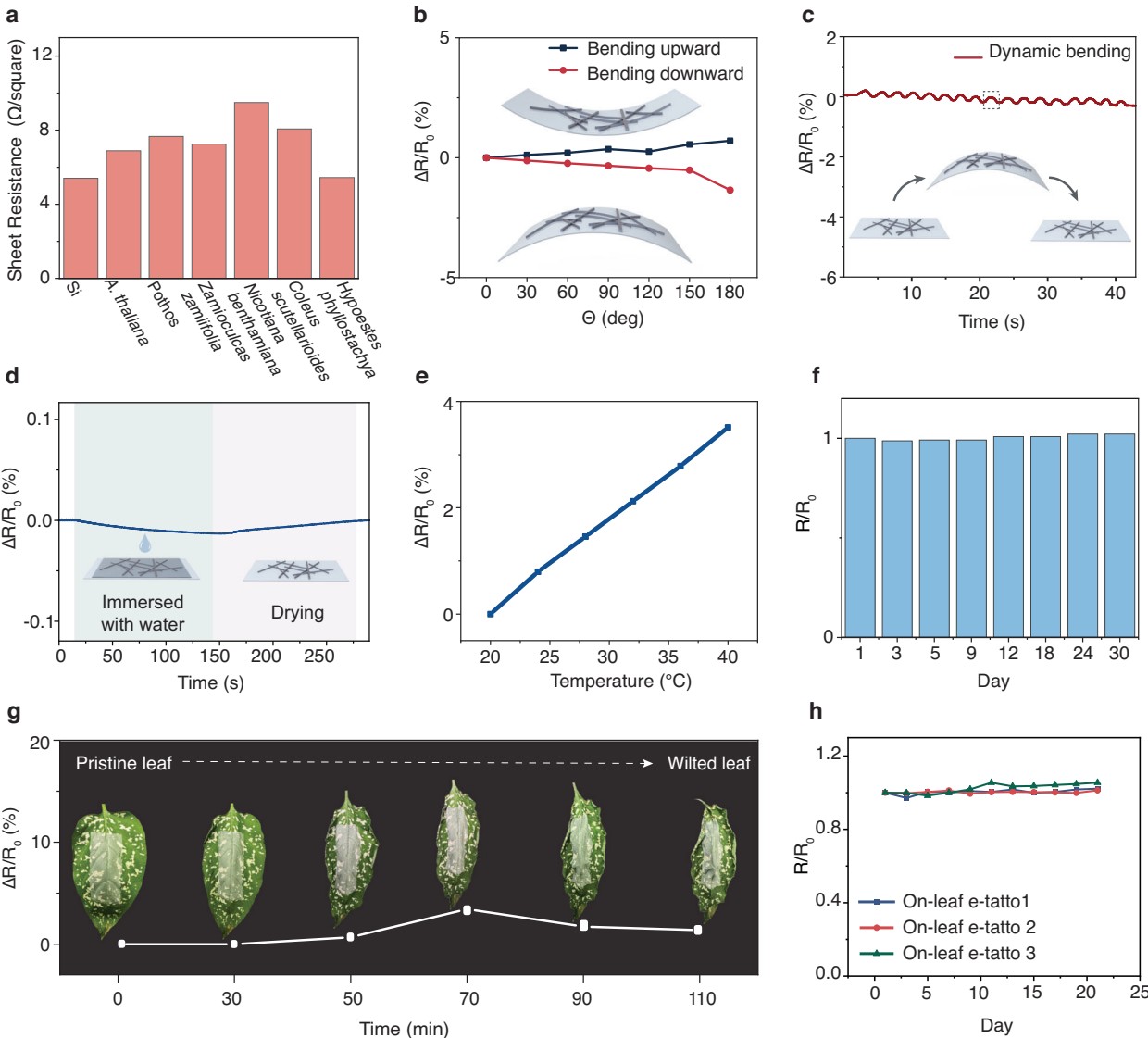

**Fig. 3 | Electrical and mechanical characterization of the e-tattoo. a** Sheet resistance of the e-tattoo transferred to different substrates. **b** Relative resistance change of the e-tattoo under various bending degrees. **c** Relative resistance change of the e-tattoo under dynamic bending (bending downward with a bending degree of 60 degrees). **d** Relative resistance change of the e-tattoo when immersed in water and dried out in the ambient condition. **e** Relative resistance change of the e-tattoo under different temperatures. **f** Relative resistance of the e-tattoo over a span of 30 days under the ambient conditions. **g** Relative resistance change of the e-tattoo transferred to a leaf undergoing a gradual dehydration process until fully dried out, accelerated in an oven at 40 °C. The inset images show photos of the leaf every 30 min, illustrating its wilting and deformation over time. **h** Relative resistance of e-tattoo on leaves over a span of 21 days. Source data are provided as a Source Data file.

## Electrochemical impedance spectroscopy of leaves via e-tattoo

Electrochemical impedance spectroscopy (EIS) is widely utilized to assess the electrical characteristics of (bio-)electrochemical systems. When a plant tissue sample is subjected to an alternating voltage input $V$, an electrical current $I$ flows through the cell walls, intercellular spaces, and plant fluids. Impedance represents the total opposition to the current flow in the plant tissues, and can be measured at multiple frequencies $f$. According to the fundamental theory of biological EIS, which involves applying an AC voltage to biological tissue through electrodes, the quality of EIS data is highly dependent on the performance of the electrodes[32]. Thus, identifying optimal electrode materials for specific biological tissues is a critical challenge in bio-EIS. Ideally, electrode materials should exhibit low impedance and maintain low impedance at the electrode-tissue interface by ensuring conformal and tight adhesion to sample tissues.

Previous plant EIS analyses primarily employed conventional metal electrodes[33,34], such as nickel (Ni), which were either attached to leaves with adhesive tape or inserted into stems. While metals offer excellent conductivity, they lack the mechanical softness and deformability of biological tissues, leading to unstable electrode-tissue interfaces. Our e-tattoo not only possesses low resistivity compared to conductive polymers but also maintains conformal attachment to various leaf surfaces due to its superior softness and thinness. Consequently, the e-tattoo emerges as an optimal electrode material for bio-EIS across diverse species, particularly for fragile and brittle plants, given its safe in-water transfer printing process.

To demonstrate the necessity of the e-tattoo for high-quality bio-EIS data, we conducted comparative EIS tests on the same site of pothos leaves (Fig. 4a-c). The e-tattoo exhibited impedance magnitude values approximately one to two orders of magnitude lower than those of three conventional metal materials in the low-frequency range (1 Hz

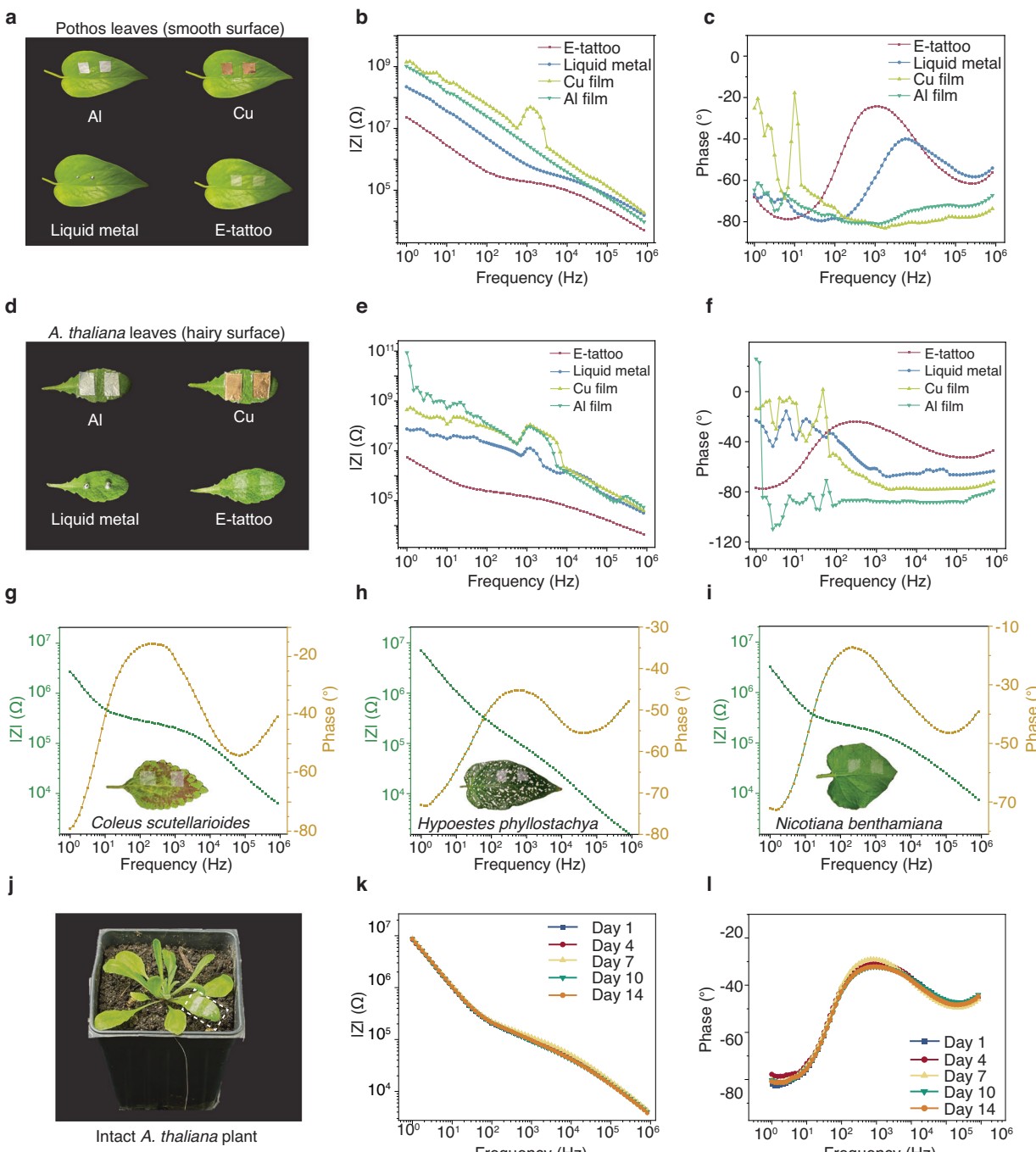

**Fig. 4 | EIS characterization of the e-tattoo compared to conventional electrodes, versatility and long-term stability tests. a** Photos of a pothos leaf with different electrodes attached to the same site. **b–c** Bode plots obtained with different electrodes on the pothos leaf. **d** Photos of the *A. thaliana* leaf with different electrodes attached to the same site. **e–f** Bode plots obtained with different electrodes on the *A. thaliana* leaf. **g–i** Bode plots acquired with e-tattoo on the leaves of *Coleus scutellarioides*, *Hypoestes phyllostachya*, and *Nicotiana benthamiana*, respectively. **j** Photo of an intact *A. thaliana* plant with one leaf transfer-printed with the e-tattoo electrodes for EIS measurements. **k–l** Bode plots collected from day 1 to day 14, taken at the same time each day. Source data are provided as a Source Data file.

to 1 kHz), indicating the lowest impedance at the electrode-tissue interface and hence the most seamless and intimate attachment to the leaf. Despite their high conductivity, metal films (Al and Cu) yielded the highest impedance values and failed to capture distinctive phase features at both high and low frequencies, implying the importance of conformal attachment for high-quality EIS data. Although liquid metal droplets show lower impedance magnitudes compared to metal films, their impedance values were still several times higher than those recorded with the e-tattoo, and the phase angle at lower frequencies

(1–10 Hz) appeared less smooth than that of e-tattoo, highlighting suboptimal interface conditions. In contrast, the e-tattoo achieved an optimal phase profile across all frequencies, further emphasizing its superior performance in delivering high-fidelity EIS data.

Obtaining high-quality EIS data from trichome-protected plants poses additional challenges as leaf surface "hairs" obstruct film attachment. Addressing this, Abdullah et al. designed a microneedle-based electrode to pierce through the leaf cuticle for EIS measurement[34]. However, the invasiveness of this approach may

potentially harm the plants. Our plant e-tattoo, owing to its superior mechanical properties and unique in-water transfer printing process, ensures conformal attachment even on trichome-covered surfaces (Supplementary Fig. 11). Figure 4d-f present EIS data of an *A. thaliana* leaf with trichomes recorded using different electrode materials. Impedance spectra obtained with metal films significantly deteriorate compared to those from Pothos leaves, as evidenced by higher impedance magnitudes and less smooth phase curves. This inferior performance is attributed to the non-glabrous surface feature of the *A. thaliana* leaves with trichomes. Although liquid metal can provide relatively smooth phase curves and lower impedance magnitudes for Pothos leaves due to its softness and deformability, its performance on *A. thaliana* leaves declines significantly potentially due to the dense coverage of trichomes. These anomalies in EIS data are primarily attributed to imperfections at the electrode-tissue interface, where non-uniform contact introduces localized impedance variations. This phenomenon, common in electrochemical measurements, can be explained by deviations from the ideal impedance response modeled by the Randles equivalent circuit[32,35]. The uneven surfaces of leaves, particularly the trichome-covered *A. thaliana* leaves, exacerbate these imperfections, resulting in more pronounced signal irregularities. These results manifest the importance of conformal attachment of electrodes for acquiring high-quality EIS data.

The printed e-tattoo provides conformal attachment even for trichome-rich surfaces, making it an optimal electrode material for bio-EIS applications. To demonstrate the e-tattoo's versatility across diverse plant species, we transfer-printed it onto leaves of *Coleus scutellarioides*, *Hypoestes phyllostachya*, *Nicotiana benthamiana*, sweet potato, and *Brassica rapa* (Fig. 4g-i, Supplementary Fig. 12), which have varying surface roughness and shapes. The corresponding impedance spectra effectively capture magnitude and phase angle variations among species, suggesting differences in cellular structure, membrane permeability, or ion transport mechanisms. The influence of the e-tattoo electrode's location and geometry on EIS data is presented in Supplementary Fig. 13, 14, demonstrating that leaf impedance measurements are reasonably tolerant to variations in electrode placement and design. Additionally, EIS data for leaves of different sizes and growth stages are provided in Supplementary Fig. 15, while EIS data for different leaves on the same plant are shown in Supplementary Fig. 16. These results highlight baseline differences in impedance across leaves, which can be attributed to the natural heterogeneity of leaf tissues, such as variations in tissue composition and ion concentration.

To validate the feasibility of long-term EIS analysis of intact plants, a 14-day experiment was conducted wherein the e-tattoo was transfer-printed onto an intact and living *A. thaliana* plant, with EIS data collected daily (Fig. 4j-l, Supplementary Fig. 17-19). The impedance spectra remained largely consistent between day 1 to day 14, with minor magnitude and phase shifts likely attributed to plant maturity and development. This stability over two weeks confirms the biocompatibility of the plant e-tattoo, particularly for the model plant *A. thaliana*. Furthermore, continuous impedance measurements over a 5-day period further demonstrate the robustness and feasibility of the e-tattoo for long-term monitoring, with measurement durations adaptable to specific experimental needs (Supplementary Fig. 20).

## Monitoring autoimmune responses in *A. thaliana*

Plant immune responses involve multi-layered signaling processes, starting from the host perception to the activation of various signaling cascades and transcriptional reprogramming[3,36]. Main players of initiating such complex signaling cascades are the plant immune receptors, many of which had been discovered as genetic elements conferring disease resistance (R) traits to the host plants[37]. These R proteins recognize pathogen invasion and trigger downstream immune responses, which include calcium influx, reactive oxygen species production, activation of mitogen-activated protein kinase

cascades, synthesis of defense hormones, and the hypersensitive response, a programmed cell death essential for plant immunity. Despite recent advances in plant immunity research, our understanding of the temporal regulation of immune signaling remains underexplored due to technical challenges. Non-invasive, real-time monitoring of immune responses has been particularly difficult to achieve. Existing analytical methodologies, such as biochemical assays, imaging techniques, and genetic analyses, fall short of providing continuous, non-invasive monitoring of immune signaling dynamics in intact plants.

To enable live monitoring of plant immune responses, we utilized a *DANGEROUS MIX* (*DM*) genetic system established in the model plant *A. thaliana*[7,38]. This system leverages paired, mismatched immune components that genetically induce defined immune responses without the need for pathogen treatment. Unlike pathogen-induced immune responses that activate multiple signaling pathways, the *DM* autoimmunity results in immune responses activated by a particular set of intracellular immune receptors[39,40]. In this study, we developed a stable *A. thaliana* transgenic line in the Mrk-0 accession, where the endogenous *DM6* immune receptor gene, encoding a canonical CC-domain containing NLR receptor RPP7, can only be activated by its partner *DM7/RPW8* from the KZ10 accession as a transgene upon ethanol induction (Fig. 1b). Neither mock treatment with water nor ethanol treatment to the wild type (WT) plants lacking the transgene induced autoimmune symptoms such as yellowing and wilting, which was evident at 4 days after ethanol induction in the transgenic line $T_3[pAlcA::RPW8.1^{KZ10}-mVenus$ #1]. This confirmed the tight regulation of the *DM6-DM7* mediated autoimmunity. All these results indicate that the inducible *DM6-DM7* autoimmunity line we established for this work is well suited for EIS monitoring using the plant e-tattoo, enabling continuous and uninterrupted tracking of immune responses with a clear onset of signaling.

To validate the feasibility of real-time and continuous immune response monitoring, the plant e-tattoos were transferred to the $5^{th}$ to $7^{th}$ leaves of each *A. thaliana* plant before treatment, which included three experimental groups as illustrated in Fig. 5a. Figure 5b presents the relative impedance magnitude of *A. thaliana* leaves in the three groups over time following induction. The data collected around 24 h post-induction (hpi) captures the full progression of immune signaling, encompassing key physiological changes such as cell death, ion leakage, and tissue remodeling. This timeline effectively reflects both the onset and resolution of the immune response, offering a comprehensive view of the underlying physiological processes. Notably, a significant drop in impedance magnitude was observed in the transgenic *A. thaliana* treated with ethanol, occurring 3–4 h post-ethanol induction. In contrast, the transgenic *A. thaliana* treated with water exhibited minimal impedance variations throughout the experimental period. Given that the leaf impedance remains stable in healthy, living plants, these results suggest a substantial physiological change in the transgenic *A. thaliana* shortly after immunity induction, well before the appearance of visual autoimmune symptoms. Meanwhile, the impedance of the WT *A. thaliana* plants remained stable throughout the experiment, despite undergoing the same induction conditions, thereby ruling out the possibility that the observed impedance drop was solely due to the ethanol exposure. Additionally, we tested the impedance changes in response to other common stressors, such as dehydration and wounding, to eliminate the possibility of these factors contributing to the observed changes (Supplementary Fig. 21, 22). Notably, water loss and wounding induced only minor upward shifts in the impedance magnitude spectrum, with small and distinct variation trends. These findings further validate that the observed impedance changes in the transgenic *A. thaliana* are specifically induced by the immune response rather than other factors.

The observed decrease in leaf impedance during the plant's immune response to ethanol exposure is primarily driven by cellular

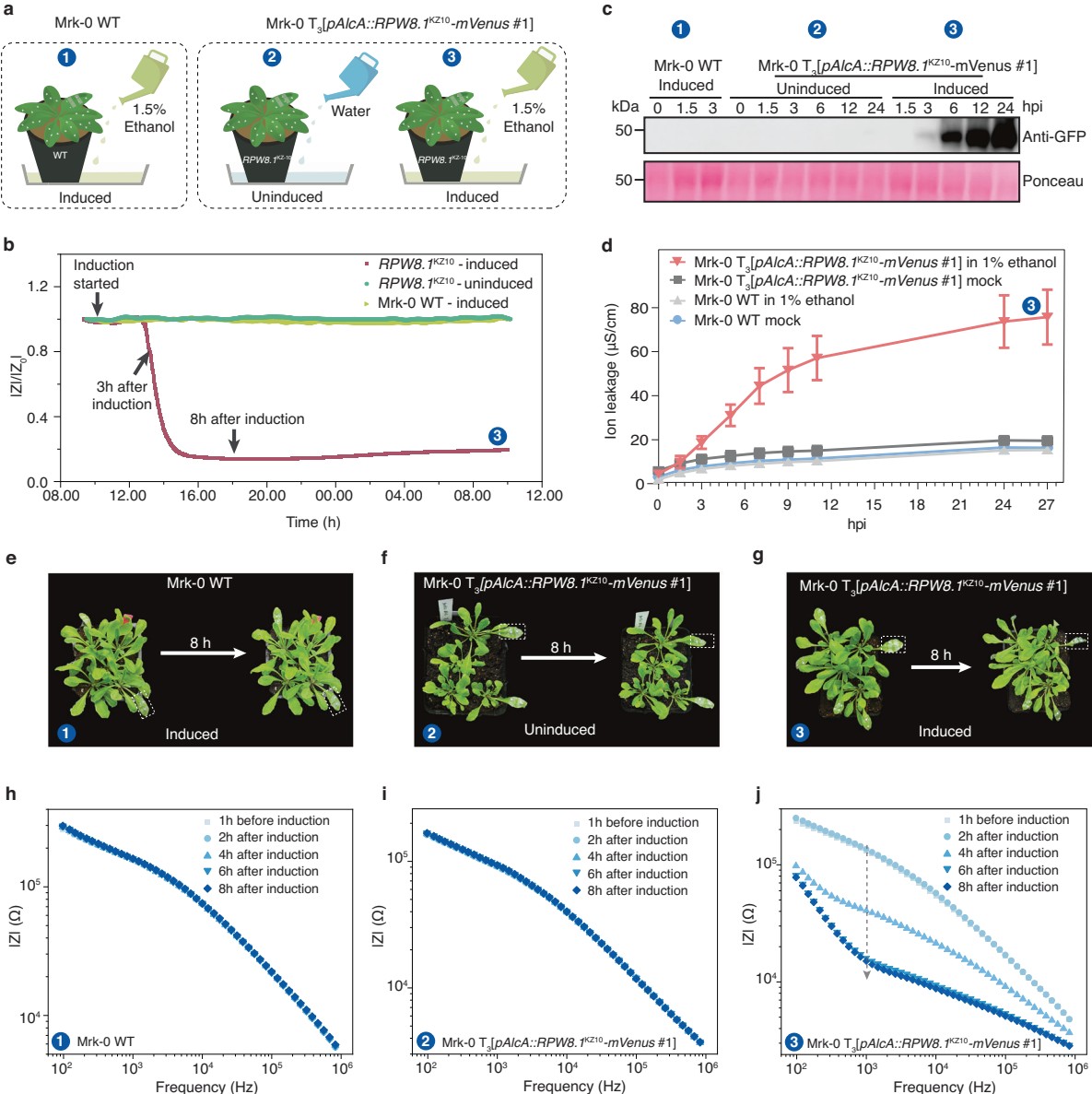

**Fig. 5 | Transgenic *A. thaliana* autoimmune response monitoring using plant e-tattoo. a** Experimental setup schematics for the control and test groups. Mrk-0 WT and uninduced transgenic plants served as negative controls. **b** Normalized impedance magnitude at 2 kHz over time of the test and control groups post induction with water or ethanol. A custom-designed portable impedance monitoring system was developed to enable continuous recording of the impedance spectra in the middle-frequency range (2 kHz–22 kHz). **c** Protein expression of ethanol-induced RPW8.1^KZ10-mVenus in 4-week-old Mrk-0 WT and transgenic plants expressing RPW8.1^KZ10-mVenus, analyzed by SDS-PAGE and western blot using an anti-GFP-HRP antibody. Similar results were obtained in more than three independent technical replicates, and a representative blot is shown. **d** Ion leakage measurements over time,

indicating hypersensitive response (HR) and cell death in ethanol-induced Mrk-0 transgenic *RPW8.1*^KZ10-mVenus plants. Leaf discs were floated in 500 mL of ddH₂O for 30 min before being transferred in sets of ten into test tubes containing 8 mL of either 1% ethanol (experimental group) or ddH₂O (mock group), with eight biological replicates per condition. The experiment was repeated in more than three independent biological replicates, yielding consistent results. Data from a representative biological replicate are shown. Data are presented as mean values ± SD. **e**–**g** Photos of control and test group plants before and after induction (at 8 hpi). **h**–**j** Impedance magnitude spectra over time of the test and control group plants. The full impedance spectrum was recorded every two hours. Source data are provided as a Source Data file.

changes associated with the activation of RPW8.1^KZ10, which activates a CNL-related immune receptor in the transgenic line. This activation triggers a hypersensitive response (HR), a well-characterized form of programmed cell death. HR involves early physiological changes, including ion leakage, loss of cell membrane integrity, and reactive oxygen species (ROS) accumulation, which collectively contribute to the decrease in leaf impedance. We confirmed the correlation between the observed impedance changes and RPW8.1^KZ10 expression by western blot analysis, detecting RPW8.1^KZ10-mVenus protein accumulation

using anti-GFP-HRP that recognizes mVenus-tagged RPW8 proteins. The RPW8.1^KZ10 proteins were detectable from 3 hpi and continued to accumulate over a 24-hour period (Fig. 5c). This observation suggests that the onset of autoimmunity starts as early as 3 hpi, which aligns well with the timing of the recorded impedance changes. These results imply a causal relationship between the onset of RPW8.1^KZ10-driven *DM6-DM7* autoimmunity and the change in leaf impedance.

To quantify the cell death triggered by RPW8.1^KZ10, we measured the electrolytic conductivity of the solution containing leaf tissues of

transgenic Mrk-0 and Mrk-0 WT. An in-tube induction assay was conducted to monitor conductivity changes over a 27-hour period. Increased ion leakage, indicative of compromised cell membrane integrity, was observed as a rise in conductivity. Conductivity increased at 3 hpi and continued to increase dramatically only in the test group: inducible Mrk-0 with RPW8.1$^{KZ10}$-mVenus in ethanol solution (Fig. 5d). This suggests severe ion leakage beginning at 3 hpi, which continued to worsen, indicating strong autoimmunity signaling mounted upon RPW8 induction to culminate in cell death. These findings correspond with the expression pattern of RPW8.1$^{KZ10}$-mVenus, further reinforcing the link between its expression and cell death. As expected, the conductivity of the control group remained stable throughout the 27-hour period, indicating preserved cell integrity and a presumably healthy physiological state. While this method provides precise quantification, it is destructive, requiring the excision of the leaf for manual measurement at each time point, potentially missing critical data during the immune response. Therefore, non-invasive, real-time continuous EIS measurement offers a valuable alternative for investigating and monitoring immune responses.

Detailed impedance spectra could provide a comprehensive assessment of the plant's physiological state during immune responses. Photos of the A. thaliana plants before and after induction are provided in Fig. 5e-g. At 8 hpi, all plants appeared similar to their pre-induction state, except for some morphological changes in a few leaves of the test group. The impedance magnitude-frequency plots for the three groups are shown in Fig. 5h-j. While the control group exhibited stable impedance spectra throughout the experiment, the test group displayed temporal impedance changes beginning at 2 hpi. These changes were most pronounced in the mid-frequency range (1 kHz to 50 kHz) compared to low and high frequencies. Between 4 hpi and 8 hpi, there was a significant decrease in impedance magnitude across all frequencies, accompanied by an alteration in the overall spectral shape, indicating pronounced cellular changes following the onset of autoimmunity signaling. These immune response-induced changes were further reflected in the phase-frequency plot, where the phase shifted from low frequency to high frequency (Supplementary Fig. 23).

The impedance change trends were validated by conducting real-time immune response monitoring in three independent trials, each utilizing a new batch of Mrk-0 plants containing the inducible transgene. The recorded EIS spectra are provided in Supplementary Fig. 24-34.Variations in immune response intensity and speed were observed among individual plants, though consistent trends in both impedance and phase data were observed across all trials. Some plants exhibited a rapid response, with significant impedance magnitude drops occurring between 2-4 hpi, while others showed a slower response with noticeable magnitude decreases after 4 hpi. Interestingly, faster responses were generally associated with stronger immune responses, as indicated by larger impedance magnitude variations at 8 hpi. These observations suggest that the induced expression of RPW8.1$^{KZ10}$ may vary among individual plants, with higher expression levels correlating with both accelerated response speeds and increased severity of cell death.

Our findings provide strong evidence that the DM6-DM7-mediated autoimmunity and associated cell death in A. thaliana specifically depend on the presence of the coiled-coil nucleotide-binding leucine-rich repeat (CNL) resistance protein RPP7 (DM6) (Supplementary Note 1 and Supplementary Fig. 35). To comprehensively analyse temporal dynamics in transcriptome associated with the activation of the immune responses, we further conducted RNA-seq on the Mrk-0 transgenic line #1 of RPW8.1$^{KZ10}$-mVenus (JW121) as well as Mrk-0 WT control in the time course ethanol treatment. Post ethanol treatment, the induced RPW8.1$^{KZ10}$ expression as well as reporter genes commonly used for immune responses exhibited rapid induction; RPW8.1$^{KZ10}$-mVenus peaked at 1.5 h, EDS16 at 3 h, and PR1 at 6 h,

suggesting a sequential activation of the salicylic acid (SA) pathway (Fig. 6a). Weighted Gene Co-expression Network Analysis (WGCNA) identified co-expressed gene modules ME1-ME5, which exhibited distinct temporal patterns (Fig. 6b). ME3, the most prominent module with 2819 genes up-regulated in ethanol-treated samples, displays a rapid induction observable as early as 3 hpi and sustained elevation afterwards until 12 hpi (Fig. 6c). This module is enriched for genes involved in SA and defense responses, a hallmark of immune responses activated by NLR immune receptors in general[41,42]. Gene Ontology (GO) enrichment analysis confirmed the ME3's association with defense processes, including responses to bacterial molecules and hypoxia (Fig. 6d). In contrast, ME5, which was down-regulated in the RPW8.1$^{KZ10}$-induced samples, is composed with 1921 genes (Fig. 6b), and its eigengene profile shows a decline post-induction, particularly within the first six hours (Fig. 6e). GO enrichment indicated involvement in photosynthesis and pigment biosynthesis as well as responses to light and auxin (Fig. 6f). To investigate 1.5% ethanol treatment as a possible stressor, WT Mrk-0 plants were treated with ethanol and assessed with RNA-seq. While ethanol application has trivial effects on transcriptome profile (Fig. 6c, Supplementary Fig. 36), it did not activate immune-related genes. Overall, the inducible DM6-DM7 system reveals a complex interplay between immune activation and growth suppression. Distinct temporal expression patterns highlight the genetic regulation of plant immunity and the trade-offs between growth and defense.

By integrating RNA-seq analysis with EIS measurements, our study establishes a direct correlation between impedance changes and gene expression dynamics during immune responses. Specifically, our time-course RNA-seq analysis of inducible DM6-DM7 in A. thaliana reveals that the rapid induction of RPW8.1$^{KZ10}$-mVenus, detected as early as 1.5 hpi and peaking at 3 hpi, coincides with initial impedance variations. This suggests that early molecular and cellular changes triggered by RPW8.1$^{KZ10}$ activation directly influence the plant's electrical properties. Furthermore, the sequential upregulation of EDS16 and PR1 aligns with sustained physiological modifications and immune responses, likely contributing to continuous impedance shifts. These findings demonstrate that impedance spectroscopy can capture dynamic immune-related physiological changes, providing a real-time, continuous, and non-invasive phenotyping tool that complements molecular and genetic approaches. The integration of plant e-tattoo and genetic approaches offers a unique platform for monitoring immune responses at both physiological and biophysical levels, and enables deeper insights into the molecular basis of plant immunity.

## Discussions

We have demonstrated a non-invasive and robust sensing technology based on e-tattoos that enables real-time and continuous plant immune response monitoring. This plant e-tattoo features an ultra-thin and substrate-free AgNW network (~100 nm), providing exceptional conformability to diverse leaf surfaces, including those with trichomes. In vivo experiments across multiple plant species confirm its superior biocompatibility and long-term attachment without adverse effects on plant health. Additionally, the e-tattoo demonstrates excellent electrical properties with low sheet resistance and stability under mechanical deformation and environmental variations, resulting in high-quality EIS data. We demonstrated long-term (>24 h) continuous EIS analysis of transgenic A. thaliana, capturing dynamic impedance spectrum variations during genetically triggered autoimmune responses. Notably, a rapid biological response (~3 h) was recorded before any visual symptoms appeared, highlighting the efficacy of this approach. RNA-seq analysis and tissue ion leakage tests further validate that EIS data accurately reflect physiological and molecular activities associated with genetically triggered autoimmune responses in A. thaliana. This epidermal e-tattoo offers a powerful tool for non-invasive and real-time plant monitoring, complementing advanced molecular and genetic techniques. We envision that this

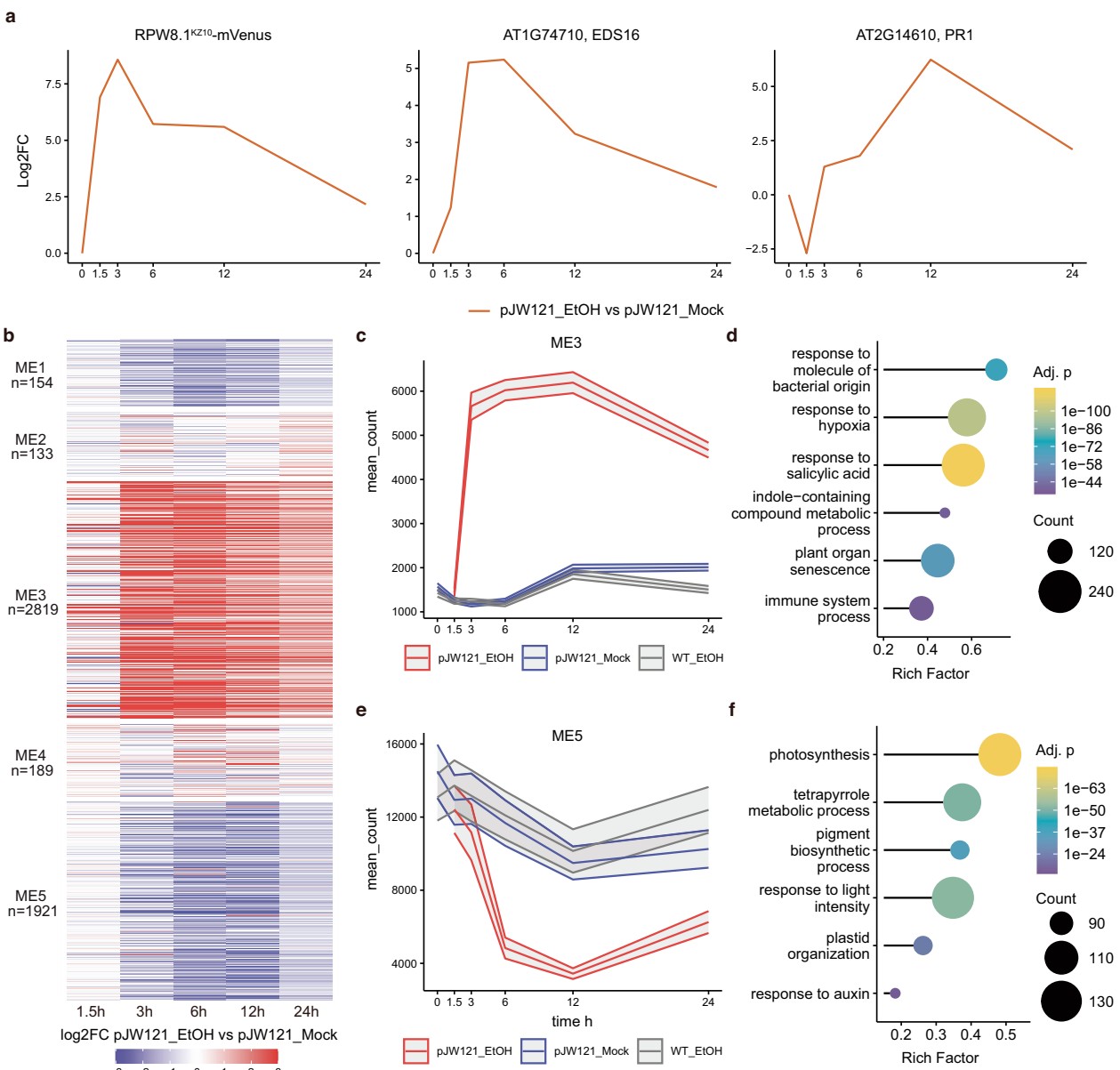

**Fig. 6 | Time-course RNA-Seq analysis of inducible *DM6-DM7* in *A. thaliana*.**
**a** Time-course log2 fold-change (log2FC) of reporter genes *RPW8.1*[KZ10]*-mVenus, EDS16*, and *PR1*. Ethanol vs. mock treatments. Analyzed using DESeq2 with Wald test. **b** WGCNA heatmap of log2FC for modules ME1-ME5 over the time course (0, 1.5, 3, 6, 12, 24 h). **c** ME3 expression profile (pJW121_EtOH in red, pJW121_mock in blue, WT_EtOH in grey), with a 90% confidence interval ribbon calculated using Student's t-distribution (two-sided, $n = 3$ biological replicates per condition). pJW121 is the construct plasmid harboring transgene *pAlcA::RPW8.1*[KZ10]*-mVenus*. **d** GO enrichment for ME3, related to defence and salicylic acid (SA) processes.

Statistical significance determined by hypergeometric test (two-sided) with *p*-values adjusted for multiple comparisons using Benjamini-Hochberg method. **e** ME5 expression profile (pJW121_EtOH in red, pJW121_mock in blue, WT_EtOH in grey), with a 90% confidence interval ribbon calculated using Student's t-distribution (two-sided, $n = 3$ biological replicates per condition). **f** GO enrichment for ME5, related to growth and photosynthesis processes. Statistical significance determined by hypergeometric test (two-sided) with *p*-values adjusted for multiple comparisons using Benjamini-Hochberg method. Simplified GO terms were generated using Wang's semantic similarity measure with a cutoff of 0.4.

technology could provide insights into the fundamental mechanisms underlying plant immunity and enhance our understanding of biological processes in stress conditions.

Further work should focus on improving the long-term durability of the on-leaf electrode and its electrical wire connection interface. One potential solution is the use of adhesive and morphing hydrogels[43], which could serve as biocompatible glues to securely anchor ultra-thin metal wires to the on-plant electrode, ensuring stable performance over extended periods. Another future direction is to further investigate the specificity of this technique. While our primary focus has been on CNL (coiled-coil NLRs) immune responses,

the adaptability of the e-tattoo enables its application to monitor other types of immune pathways. For instance, we demonstrated its capability by tracking the EIS changes in *A. thaliana* with induced TNL (Toll/interleukin-1 receptor-like NLRs) immune responses. Notably, the impedance signatures of these two immune pathways exhibit distinct characteristics, including differences in magnitude, direction, and rate of variation, aligning well with their respective signaling kinetics (Supplementary Fig. 37-39 and Supplementary Note 2). These results reinforce the potential of plant e-tattoo as a robust, non-invasive, and continuous monitoring tool for diverse immune responses.

Beyond differentiating specific NLR-mediated responses, future work could focus on optimizing the e-tattoo's sensing capability to differentiate between pattern-triggered immunity (PTI) and effector-triggered immunity (ETI). PTI, which is initiated by pattern-recognition receptors (PRRs) at the cell surface, generally leads to more transient and moderate physiological changes than ETI[6,44,45]. Given that PTI responses involve ROS bursts, cytosolic calcium elevation, and callose deposition[46], all of which can affect the electrical properties of plant tissues, we hypothesize that EIS monitoring with e-tattoos could potentially capture these dynamic changes. However, detecting PTI effectively remains a challenge due to the relatively weaker and shorter-lived nature of PTI responses, which may lead to subtle impedance variations. Enhancing sensitivity will require optimization in several key areas, including pathogen-associated molecular pattern (PAMP) treatment strategies, electrode design and placement, measurement protocols, and signal processing techniques. Addressing these challenges could improve the e-tattoo's ability to distinguish between PTI and ETI, further expanding its applicability in plant immunity research.

While this study establishes a rigorous framework for characterizing NLR-specific impedance signatures, it primarily serves as a proof of concept, laying the groundwork for future research applicable to a broader range of immune responses and stressors. An important future direction involves investigating the e-tattoo's performance in plants infected with virulent and avirulent pathogens. Such studies would further validate its practical application in monitoring biotic stress responses and broaden the scope of its use in plant immunity research. Additionally, the integration of plant e-tattoo and genetic approaches could offer a unique platform for monitoring immune responses at both physiological and biophysical levels, enabling deeper insights into the molecular basis of plant immunity. To enhance the specificity of this technique, machine learning could play a pivotal role in differentiating multiple stressors or immune responses by analyzing complex EIS patterns. Advanced algorithms could extract nuanced features from impedance data, enabling more precise characterization of plant responses under varying conditions. This synergy between the plant e-tattoo and machine learning presents a promising pathway for developing a high-throughput, intelligent platform for studying plant immunity and stress physiology.

## Methods

### Materials
AgNWs was purchased from Merck Pte Ltd (diam. × L 100 nm × 6 µm, 0.5% (isopropyl alcohol suspension)). Graphene oxide water dispersion was purchased from Graphenea (0.4 wt% Concentration). *Arabidopsis thaliana* accession Mrk-0 (ABRC stock number: CS1374), Col-0 (ABRC stock number: CS70000) and Cdm-0 seeds (ABRC stock number: CS76410) are from lab stock.

### Fabrication of the e-tattoo
The fabrication of the plant e-tattoo involves two primary processes: vacuum filtration for film formation and in-water transfer, as detailed in Supplementary Fig. 1. To begin, prepare the AgNW suspension by diluting 30 µL of the as-purchased AgNW suspension in 300 mL of pure water. Ultrasonicate the mixture for 2 min to achieve a uniform dispersion of the AgNWs. Next, set up the vacuum filtration apparatus by placing an anodic aluminum oxide (AAO) film on the filtering head. Clean the filtering cup with isopropyl alcohol (IPA) and rinse it with deionized (DI) water before securing it with a clamp. To facilitate the detachment of the AgNW film from the AAO membrane after filtration, a thin layer of graphene oxide (GO) is introduced as an interfacial layer. First, prepare the GO suspension by diluting 50 µL of a GO dispersion (which is diluted 1:20 with DI water) in 250 mL of pure water. Stir the GO suspension for 1 minute to ensure uniform distribution. Pour the GO suspension into the filtering cup and initiate filtration by turning on

the vacuum pump at a pressure of 24 mbar. When 20–30 mL of the GO suspension remains in the filtering cup, carefully add the AgNW suspension and continue the filtration process until all the water has evaporated. The introduced GO on AAO film facilitates the release of the AgNW film from the AAO membrane, allowing the film to float on the water surface for subsequent transfer printing. The GO layer remains beneath the AgNW layer after transfer printing.

To exfoliate and transfer the AgNW film, prepare a container filled with water and place the AgNW/AAO film on the water's surface. Apply gentle pressure to the edge of the AAO film to allow water to seep between the AgNW and AAO film, utilizing water surface tension to detach the AgNW film. The AAO film will sink to the bottom of the container, while the AgNW film will float on the surface. To transfer the AgNW film, submerge leaves or other substrates below the water surface and gently lift them to pick up the AgNW film. As the water evaporates at ambient temperature, the AgNW film will adhere firmly to the substrates, forming the e-tattoo for further testing. For patterning the AgNW-based e-tattoo, use a polyimide shadow mask with laser-cut designs, which should be attached to the leaf surface before transferring the AgNW film. After positioning the AgNW film on the leaf and allowing it to dry, carefully peel off the shadow mask. This will leave behind the desired patterns on the leaf surface. For all EIS analyses, unless otherwise specified, a two-square-electrode arrangement is used for collecting EIS data from leaves. A small droplet of liquid metal is applied to the e-tattoo electrode to establish an electrical connection with the commercial copper wire (50 µm in diameter) used for testing.

### Characterization of the e-tattoo
The thickness of the developed sensor was measured by the surface profile (Bruker, DektakXT). Sheet resistance was measured by a four-probe measurement system (AiT, CMT-SR2000N). Photographs and videos were acquired by an iPhone 14 camera. The resistance of the e-tatoo under different bending conditions was measured by the programmable electrometer (Keithley, 6514), and an oscilloscope (Agilent, InfiniiVision, DSO-X3034A) was connected to the electrometer using a bayonet nut coupling (BNC) cable for real-time data acquisition. The temperature variations were created and controlled using a hot plate.

### Leaf fluorescence image
The fluorescence image was detected using the Bio-Rad ChemiDoc MP Imaging System. The fluorescence signal was detected using excitation and emission wavelengths specific to the Alexa Fluor 488 channel, with excitation at approximately 488 nm and emission at approximately 519 nm. For quantification, grayscale fluorescence images were first acquired from the Bio-Rad ChemiDoc MP Imaging System, with pixel intensity values ranging from 0 (black) to 255 (white), corresponding to fluorescence intensity. These grayscale images were then processed using ImageJ, where a false-color scale was applied to map intensity values to colors, enhancing the visual interpretation of the data.

### EIS data collection
The impedance $Z(\omega)$, where $\omega = 2\pi f$, is a complex value that relates to the voltage and current as

$$Z(\omega) = \frac{V(\omega)}{I(\omega)} = |Z(\omega)|(\cos(\phi(\omega)) + j\sin(\phi((\omega)))) \quad (1)$$

where $|Z(\omega)|$ represents the magnitude of the impedance, and $\Phi(\omega)$ is the phase difference between the voltage and the current. The full impedance spectrum was measured across a frequency range of 1 Hz to 1 MHz using an electrochemical workstation (CHI600E). The voltage of the alternating electrical field is set to 0.7 V. For real-time and continuous tracking of the impedance spectrum, a custom portable system was developed. This system includes an AD5933 impedance

converter board connected to an Arduino Mega 2560, enabling real-time impedance acquisition with a frequency range of 2 kHz to 22 kHz.

## Plant growth conditions

*A. thaliana* Mrk-0, Col-0 and Cdm-0 seeds were surface sterilized with 70% ethanol and 0.5% Triton X-100 and then stratified in the dark at 4°C for 6 days before planting on the soil or ½ MS agar plate. Plants were grown on soil at 22 °C at 60% relative humidity under long day condition (16/8 h light/dark) with fluorescent and incandescent lamp light of 125-175 µmol m$^{-2}$ s$^{-1}$ Percival chamber (Model AR-41L3, Percival Scientific, Perry, Iowa, United States).

## Constructs and transgenic plants

The ethanol-inducible binary vector pDH870 was first generated by inserting the *AlcA* and *AlcR* elements into pGREEN-IIS_41012 by Gibson Assembly (New England Biolabs, NEB). The genomic fragment of *RPW8.1*$^{KZ10}$ fused to mVenus in the frame at its C-terminus was cloned in pDH870 by Gibson Assembly to generate the final binary construct. This was transformed into Mrk-0 via the floral dipping method. The transformed plants were selected using hygromycin (Thermo Fisher) selection on ½ MS agar plate.

## Ethanol-induced immune response in transgenic *A. thaliana*

To establish a well-controlled immune response, we prepared *A. thaliana* Mrk-0 wild type (WT) and Mrk-0 carrying the inducible *DM6-DM7* transgene (Mrk-0 T$_3$[*pAlcA::RPW8.1*$^{KZ10}$-*mVenus* (construct name: *JW121*) #1]), whose immune responses can be triggered by 1.5% ethanol treatment. Three groups of Mrk-0 pots were prepared for the experiment: Mrk-0 WT with ethanol treatment, transgenic Mrk-0 T$_3$[*pAlcA::RPW8.1*$^{KZ10}$-*mVenus* #1] with water treatment (mock), and transgenic Mrk-0 T$_3$[*pAlcA::RPW8.1*$^{KZ10}$-*mVenus* #1] with ethanol treatment. We hypothesized that immune responses would be effectively induced solely in the transgenic Mrk-0 plants exposed to ethanol, with the other two groups serving as controls.

For the ethanol induction process, five-week-old transgenic Mrk-0 plants (third generation) were irrigated with 1.5% ethanol (experimental group) or tap water (control group) while covered with a transparent plastic dome. The dome was maintained throughout the sampling and impedance measurement period.

## Protein extraction and western blot assay

To detect the induced expression of RPW8.1$^{KZ10}$-mVenus in *A. thaliana*, 100 mg plant samples at different induced time points (0, 1.5, 3, 6, 12, and 24 hpi) were harvested. Total protein was extracted with 100 µL TBS extraction buffer (50 mM Tris-HCl [pH 7.5], 150 mM NaCl, 1 mM EDTA [pH 8.0], 0.4% Triton X-100, proteinase inhibitor cocktail). Proteins were separated by 12% SDS-PAGE and detected by immunoblot using anti-GFP-HRP antibody (130-091-833, Miltenyi Biotec, Germany).

## Ion leakage conductivity assay

To measure the ion leakage of inducible autoimmune *A. thaliana* plants, two to three discs of the leaves from the 5$^h$ to 7$^{th}$ position from five-week-old plants were punch out using 8 mm biopsy punch (Med-Blades). Discs were floated on 500 mL ddH$_2$O for 30 min, and then every 10 discs were transferred into test tube with 8 mL 1% ethanol (for the experimental group) or ddH$_2$O (for the control group) with eight replicates. Conductivity was measured using Orion Conductivity Meter (Thermo Scientific, Beverly, MA, USA).

## RNA extraction and sequencing

To investigate the transcriptional response of inducible *DM6-DM7*, Mrk-0 *pJW121* (*pAlcA::RPW8.1*$^{KZ10}$-*mVenus*) Mock and Mrk-0 *pJW121* (*pAlcA::RPW8.1*$^{KZ10}$-*mVenus*) ethanol treated seedlings were collected at various time points (0, 1.5, 3, 6, 12, 24 h). Shoots of treated plants were carefully cut with a blade, collected, and immediately frozen in liquid nitrogen. Samples were ground and homogenised using a TissueLyser II (QIAGEN). Total RNA was extracted using the LogSpin method[47]. Each time point had three biological replicates for both ethanol and mock treatments.

The extracted RNA was sent to Novogene for quality control, strand-specific library preparation, and paired-end sequencing on the Illumina NovaSeq X Plus platform, generating approximately 30 million reads per sample.

## mRNA-seq analysis

The genome (TAIR10) and gene annotations from Ensembl were manually modified to include the insert transgene sequences. Reads were mapped to the modified genome using Hisat2[48]. Samtools[49] was used to sort the reads, and featureCounts[50] was used to assign reads to genomic features. Differential expression analysis was performed using DEseq2[51].

Weighted correlation network analysis (WGCNA) was conducted following the protocol by Langfelder and Horvath[52] using a signed network. A subset of 20% of the most variable genes in the genome was used for the analysis. Gene Ontology (GO) analysis was performed using clusterProfiler[53]. Visualisations were primarily created using ggplot2[54].

## Reporting summary

Further information on research design is available in the Nature Portfolio Reporting Summary linked to this article.

## Data availability

The RNA-Seq data generated in this study have been deposited in the NCBI SRA database under accession code PRJNA1080050. Source data are provided with this paper.

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

## Acknowledgements

This work was supported by Reimagine Research Scheme projects, National University of Singapore, A-0009037-03-00 and A-0009454-01-00 (to C.L.); and A-0004772-00-00 and A-0004772-01-00 (to E.C.); RIE 2025 – Industry Alignment Fund – Industry Collaboration Projects (IAF-ICP) (Grant I2301E0027) (to C.L.); National Research Foundation Singapore grant, CRP28-2022-0038 (to C.L.); and the National Natural Science Foundation of China (Grant No. 62401372) (to T. H.).

## Author contributions

T.H., J.W., and E C. conceived the idea. T.H. and C.L. planned the research. T.H. developed the electronic tattoo and conducted its characterization. J.W. developed plant materials. T.H., J.W., and Y.Y. conducted the on-plant experiments. D.H. performed the molecular bioinformatic analysis. T.H. wrote the paper with input from all the authors. E.C. and C.L. supervised this project.

## Competing interests

The authors declare no competing interests.
