## [Peer Review file · Nature Communications]

Epidermal electronic-tattoo for plant immune response monitoring

Corresponding Author: Professor Chengkuo Lee

Version 0:

Reviewer comments:

Reviewer #1

(Remarks to the Author)

In this manuscript, the authors reported an ultra-thin electronic tattoo patch made of silver nanowire (Ag NW) films for plant immune response monitoring. The patch measures electrical impedance (EIS) signals in real-time by applying alternative voltage. The attachment and biocompatibility of the patch were investigated by visual inspection of various plants wearing the patches for up to 60 days. For a proof-of-concept application demonstration, a transgenic *Arabidopsis thaliana* plant that has an immune response to ethanol was tested. The sensor patch was able to capture detectable plant immune responses within 3-5 hours of ethanol stress. The sensor results were validated by conventional molecular analysis such as RNA-seq and tissue ion leakage tests.

The main innovation of the study is the development of an in-water transfer printing protocol that allows direct attachment of a thin layer of Ag NW electrode (<100 nm thin) on rough plant leaves without the need for any substrate support. This not only improves the contact between the electrode and plant tissues, which enables robust EIS analysis, but also improves the flexibility and stretchability of the sensor patch. However, the fabrication method does not address other parts of concerns, such as the stability of the Ag NW layers. Will the attached Ag NW network be self-robust enough in the long term to survive in complex plant growth environments, such as remaining attached and intact during raindrops? How to deal with the chemical stability of Ag NWs from oxidation? This study used liquid metal droplets as the connectors between the sensor patch and the conductive wires, which does not seem a long-term solution. What will be a more robust connection protocol to power the electrode and transmit the data?

More fundamentally, several on-plant EIS measurement studies have been demonstrated before (ref. #26, 28, and others). Many studies focus on applying different stresses and acquiring the subsequent signals. While it is an essential first step for sensor development, for real applications, the opposite pathway (sensor signal-to-stress conversion) should be more considered. A simple question is how to make the EIS signals more specific. In the real field, once an EIS signal is detected from the plants, what does it mean? Does it always mean ethanol stress (obviously not)? Without addressing this issue, the study would be another technical demonstration but lacks a clear path for real usability.

Overall, the study describes some improvements in sensor device fabrication, but the sensing mechanism itself has been previously demonstrated and remains questionable for real applications. Other detailed comments are listed below:

Detailed comments:

1. During the transfer printing process, it seems that graphene oxide (GO) was used as the interfacing layer between Ag NWs and aluminum oxide (AAO). After the transfer printing, does GO remain on Ag NWs or on AAO? What is the vacuum pressure used to form the Ag NW film? More experimental data is needed.
2. In Fig S3, it seems that a mask could be used to create different patterns for the e-tattoo on the leaf. One question is that since the Ag NW film deposited on the mask is continuous, how could the peel-off action of the mask early break up the Ag NW film and leave a pattern behind. If the Ag NW film can be easily broken, does it imply the film might be fragile to mechanical damage?
3. For the repeatability of leaf impedance measurement, would the location and geometry of the Ag NW electrodes matter?

For instance, how far the two electrodes should be spaced on the leaf? Do different plant tissues (young vs. old leaves) affect the baseline impedance value?

4. For the biocompatibility test (Fig 2a), both plants with or without the tattoo patch seem to be under stress. Several leaves turned yellow after 7 days of growth. Any explanation? For the 60-day test, are photos showing the same plant? The container has obviously been changed.

5. In Fig 2d, does the SEM really show a stoma? More SEM images are needed to prove the opening is indeed a stoma.

6. What is the transparency of the Ag NW film? A UV-vis absorption spectrum of the Ag NW film should be provided.

7. In Fig 2f, where does the fluorescence come from? Is the fluorescence from leaf chlorophyll? What are the excitation and emission wavelengths?

8. In Fig 4b and 4e, there are signal bumps in several impedance spectra at the frequency of around 10^3 Hz. What is the cause of that?

9. During the immune response of the plant to ethanol exposure, what actually contributed to the decrease in leaf impedance value? Is it related to plant cell death and increased conductivity? More clear explanation of direct factors that lead to leaf impedance decrease is needed.

10. What is the voltage of the alternating electrical field?

11. The conclusion paragraph mentioned that the sensor could detect plant biological response within ~3 hours, while the abstract mentioned it was within 5 hours. This inconsistency needs to be corrected.

Reviewer #2

(Remarks to the Author)

Reviewer #3

(Remarks to the Author)

In this manuscript, He et al. described a silver nanowire-based, substrate-free electrode for monitoring the health of plants using impedance spectroscopy. The authors demonstrated the attachment of the e-tattoo onto the leaves of Arabidopsis to measure their electrochemical impedance under different immune conditions. While continuous, non-invasive monitoring of plant health is quite interesting, there are several major issues that require careful attention:

The manuscript lacks a clear explanation of the reasoning behind impedance changes in plant leaves. From the images presented in Figure 1, it seems that the reference electrode (REF) and working electrode (WE) are both on the same leaf and adjacent to each other. The observed impedance changes could simply reflect the hydration state of the leaf, which might correlate with the immune response. What is the main mechanism driving these impedance changes?

Although the authors achieved conformable contact with the leaf, it remains unclear how a long-term, reliable ohmic contact is maintained with the e-tattoo itself. Demonstrating continuous longitudinal measurements would be crucial to highlight the advantages of the e-tattoo.

The manuscript lacks sufficient statistical analysis. The group size, metrics for comparison, and the significance of the biocompatibility tests are unclear. Is biocompatibility solely assessed through images of leaves with and without the e-tattoo?

The impedance measurements are performed on only one leaf. Would this single-leaf approach be representative of the overall plant health? Additionally, what is the variability in impedance across different leaves of the same plant?

How did the authors quantify the color bar in Figure 2F? It is unclear how the fluorescence image was colorized and at what wavelength it was captured.

A major advantage of substrate-free electrodes is their potential to stretch and grow with the leaf. Given the capability of the existing electrodes to perform electrochemical impedance spectroscopy (EIS), it is critical to demonstrate the added value of these electrodes. Their stability and measurement capability at different stages of growth should be validated. Furthermore, is a 24-hour timeline physiologically relevant for such plants?

Reviewer #4

(Remarks to the Author)

In the manuscript entitled "Epidermal electronic-tattoo for plant immune response monitoring," He et al. report an ultrathin, substrate-free, and highly conductive electronic tattoo (e-tattoo) designed for plants, enabling immune response monitoring through non-invasive and robust electrical impedance spectroscopy (EIS) analysis. The authors demonstrate continuous EIS analysis of live transgenic plants for over 24 hours, capturing the onset of NLR-mediated acute immune responses within five hours post-induction, even before visible symptoms appear. This represents a promising technical advancement with great potential to enhance our understanding of plant immune activation mechanisms. However, the manuscript currently lacks evidence to confirm whether the e-tattoo specifically detects NLR-mediated immune responses. Furthermore, for its broader application in plant immunity research, it is essential to show whether the e-tattoo can monitor physiological and molecular changes in various plants upon exposure to living virulent and avirulent pathogens. Without this validation, the scope of the e-tattoo's application would be quite limited. Additionally, the manuscript should discuss how this technology could be integrated with advanced molecular and genetic approaches to advance plant immunity research. Addressing these points would significantly strengthen the impact of this invention.

Major Points

Please provide evidence that the e-tattoo can specifically detect NLR-mediated immune responses. Experiments involving MAMP or DAMP treatments and their effect on EIS analysis would help clarify this point. Testing the response to abiotic stresses, such as wounding or dehydration, would also demonstrate the specificity and robustness of the technology. Please test the e-tattoo's ability to detect NLR-mediated immune responses in plants infected with living virulent and avirulent pathogens. This would validate its practical application in studying immune responses to biotic stress and help determine the specificity of EIS signals for NLR-mediated responses. I would also suggest testing eTattoo on other plants (closer to crop species), such as *Nicotiana benthamiana* and tomato with virulence/avirulence pathogens.

Version 1:

Reviewer comments:

Reviewer #1

(Remarks to the Author)

In this revised manuscript, the authors did a great job on responding to the previous questions. The comments are properly addressed by either providing more explanations (many of them have actually been incorporated into the main manuscript, rather than a simple reply in the response letter) and extensive new experiments to confirm the results. There are no further comments from this reviewer.

Reviewer #2

(Remarks to the Author)

Reviewer #3

(Remarks to the Author)

Authors were able to address all my comments and questions.

Reviewer #4

(Remarks to the Author)

The manuscript has been extensively improved, and I appreciate the authors' efforts in addressing previous concerns. Before publication in Nature Communications, I recommend adding two additional datasets to further strengthen the study.

Firstly, the authors have clearly demonstrated that TNL- and CNL-mediated immune responses exhibit distinct impedance signatures. To complement this, please provide data showing when cell death is induced upon ethanol activation of DM10 in the Cdm-0 background.

Secondly, as the authors noted in their reply, plant pathogens trigger both PTI and ETI. It is important to assess whether e-tattoo technology can differentiate these two immune responses, demonstrating its specificity and potential application in plant immunity research. While the authors have tested the effect of wounding on impedance signatures, I recommend additional experiments using flg22 or other well-characterized PAMPs to determine whether this method can reliably distinguish PTI from ETI.

Version 2:

Reviewer comments:

Reviewer #4

(Remarks to the Author)

The authors have satisfactorily addressed all the points I raised, and the manuscript is now ready for publication in Nature Communications.

To the reviewers,

We thank all the reviewers for their insightful and constructive comments on our manuscript. Below, please find a point-to-point response to the comments. As detailed below, we have extensively revised the main text to address the concerns raised, which are highlighted in blue in the revised manuscript. We believe that the manuscript is much improved as a result.

Reviewer #1 (Remarks to the Author):

In this manuscript, the authors reported an ultra-thin electronic tattoo patch made of silver nanowire (Ag NW) films for plant immune response monitoring. The patch measures electrical impedance (EIS) signals in real-time by applying alternative voltage. The attachment and biocompatibility of the patch were investigated by visual inspection of various plants wearing the patches for up to 60 days. For a proof-of-concept application demonstration, a transgenic Arabidopsis thalian plant that has an immune response to ethanol was tested. The sensor patch was able to capture detectable plant immune responses within 3-5 hours of ethanol stress. The sensor results were validated by conventional molecular analysis such as RNA-seq and tissue ion leakage tests.

The main innovation of the study is the development of an in-water transfer printing protocol that allows direct attachment of a thin layer of Ag NW electrode (<100 nm thin) on rough plant leaves without the need for any substrate support. This not only improves the contact between the electrode and plant tissues, which enables robust EIS analysis, but also improves the flexibility and stretchability of the sensor patch. However, the fabrication method does not address other parts of concerns, such as the stability of the Ag NW layers. Will the attached Ag NW network be self-robust enough in the long term to survive in complex plant growth environments, such as remaining attached and intact during raindrops? How to deal with the chemical stability of Ag NWs from oxidation? This study used liquid metal droplets as the connectors between the sensor patch and the conductive wires, which does not seem a long-term solution. What will be a more robust connection protocol to power the electrode and transmit the data?

Reply:

We thank the reviewers for their thoughtful comments and for highlighting the novelty of our work. We agree that stability of the AgNW layer is important for long-term monitoring. To address this concern, we have evaluated the long-term stability of the Ag NW layer over a one-month period, as shown in Fig. 3f. Additionally, the impact of water droplets, including simulated raindrop conditions, on the integrity of the Ag NW layer has been tested, with results presented in Fig. 3d. These studies demonstrate the capability of our e-tattoo to maintain performance over long-term monitoring. To further assess the robustness of the AgNW layer in realistic plant environments, we conducted additional experiments to evaluate its impedance performance on *A. thaliana* (*Arabidopsis thaliana*) over a three-week period. We have included these results in a newly added Fig. 3h and revised the description of the e-tattoo's characterization in the Results section (page 6) accordingly:

“To further assess the robustness of the plant e-tattoo in real-world plant environments, we monitored its resistance changes on actual leaves over a three-week period (Fig. 3h). The results revealed minimal variations throughout this timeframe, further validating the e-tattoo's stability and reliability.”

Fig. 3h, Relative resistance of e-tattoo on leaves over a span of 21 days.

Regarding the connectors between the sensor patches and the conductive wires, liquid metal droplets were selected for this study due to their high conductivity and ease of application. In the future applications, hydrogel could be a potential alternative due to their flexibility, adhesiveness, and biocompatibility. To address this, we have included the following future prospect in Discussion section (page 12):

“Further work should focus on improving the long-term durability of the on-leaf electrode and its electrical wire connection interface. One potential solution is the use of adhesive and morphing hydrogels⁴⁴, which could serve as biocompatible glues to securely anchor ultra-thin metal wires to the on-plant electrode, ensuring stable performance over extended periods.”

More fundamentally, several on-plant EIS measurement studies have been demonstrated before (ref. #26, 28, and others). Many studies focus on applying different stresses and acquiring the subsequent signals. While it is an essential first step for sensor development, for real

applications, the opposite pathway (sensor signal-to-stress conversion) should be more considered. A simple question is how to make the EIS signals more specific. In the real field, once an EIS signal is detected from the plants, what does it mean? Does it always mean ethanol stress (obviously not)? Without addressing this issue, the study would be another technical demonstration but lacks a clear path for real usability.

Reply:

We thank the reviewers for their insightful comments. We acknowledge that plants are exposed to diverse biotic and abiotic stresses, and a detected EIS signal could originate from various sources, requiring improved specificity for practical usability.

In our study, ethanol was used as a model stimulus to trigger a well-characterized immune response. This allowed us to evaluate the sensitivity and capability of our EIS technique to detect physiological changes associated with immune activation. Unlike previous studies (e.g., refs. #26, 28), which often involved detached leaves or temporary measurements, our work enabled continuous, non-invasive EIS monitoring on intact and fragile leaves, including *A. thaliana*. This was made possible by the unique advantages of our plant e-tattoo electrodes.

To address the reviewer's concern regarding signal specificity, we conducted additional experiments comparing immune responses with other types of stresses, such as drought and wounding. The results, now presented in a new Supplementary Fig. 21, 22, demonstrate that immune responses induce more pronounced impedance changes, due to the significant cellular activity associated with immune signaling. Furthermore, we performed experiments to monitor EIS signals during different types of immune responses (CNL- and TNL-mediated immune response), with the results summarized in newly added Supplementary Fig. 37-39 and Supplementary Note 2. These data illustrate distinct impedance variations associated with immune activities, highlighting the potential for distinguishing between different stressors.

Supplementary Fig. 21. EIS response of plant leaves to dehydration. a, Impedance magnitude spectra of a *A. thaliana* leaf at different levels of LWC (loss of water content). **b,** Phase spectra of a *A. thaliana* leaf at different levels of LWC.

Supplementary Fig. 22. EIS response of plant leaves to wounding. The wound was created by making a small hole with a tweezer at the midpoint between the two e-tattoo electrodes. EIS measurements were taken 10 minutes post-wounding to assess the response. **a-b**, Impedance magnitude and phase spectra of a sweet potato leaf before and after wounding. **c-d**, Impedance magnitude and phase spectra of another sweet potato leaf before and after wounding with two cuts.

Supplementary Fig. 37. Monitoring TNL-mediated immune response in transgenic *A. thaliana* (Cdm-0) using a plant e-tattoo. **a**, Images of *A. thaliana* before and after ethanol induction to activate the DM10^{TueScha-9} TNL¹³ in the Cdm-0 background. Four-week-old transgenic Cdm-0 plants carrying pAlcA::DM10^{TueScha-9} were treated with 1% ethanol via water irrigation for seven days under a covered dome. Tissue necrosis became visible after 4 days post-induction (dpi), with the lower image showing

necrosis progression at 4 dpi. **b**, Normalized impedance magnitude at 2 kHz over time for the test and control groups following induction with ethanol or water.

Supplementary Fig. 38. Impedance signature of CNL- and TNL- mediated immune response. a, Normalized impedance magnitude at 2 kHz of a typical *A. thaliana* (Cdm-0 T₃ pAlcA::DM10^{TueScha-9}) post induction with ethanol. **b-c**, The corresponding bode plots of the control and test plants collected at time points indicated with black and red triangular in **a**. **d**, Normalized impedance magnitude at 2 kHz of a typical *A. thaliana* (Mrk-0 T₃ pAlcA::RPW8.1^{KZ10}) post induction with ethanol. **e-f**, The corresponding bode plots of the control and test plants collected at time points indicated with black and red triangular in **d**.

We have also expanded the Discussion section (page 12-13) to discuss the signal specificity and potential future directions. These include exploring advanced data analysis techniques, such as machine learning, to correlate specific impedance patterns with distinct stress types.

“While our current study provides a rigorous and controlled framework for characterizing NLR-specific impedance signatures, this work serves as a proof of concept, laying the foundation for future studies applicable to a broader range of immune responses and stressors. An important future direction involves investigating the e-tattoo’s performance in plants infected with virulent and avirulent pathogens. Such studies would further validate its practical application in monitoring biotic stress responses and broaden the scope of its use in plant immunity research. Additionally, the integration of plant e-tattoo and genetic approaches could offer a novel platform for monitoring immune responses at both physiological and biophysical levels, and enables deeper insights into the molecular basis of plant immunity. To enhance the specificity of this technique, machine learning could play a pivotal role in differentiating multiple stressors or immune responses by analyzing complex EIS patterns. Advanced algorithms could extract nuanced features from impedance data, enabling more precise characterization of plant responses under varying conditions. This synergy between the plant e-tattoo and machine learning presents a promising pathway for developing a high-throughput,

intelligent platform for studying plant immunity and stress physiology.”

Overall, the study describes some improvements in sensor device fabrication, but the sensing mechanism itself has been previously demonstrated and remains questionable for real applications. Other detailed comments are listed below:

Reply:

We thank the reviewer for noting the novelty of our work in sensor device fabrication. We agree with reviewer that the concept of EIS sensing is well-known, but that the design of e-tattoo capable of monitoring plant immune response is new. Current sensing methods of immune response are predominantly limited by intermittent measurements, such as destructive analyses conducted in laboratories. These approaches often suffer from low spatial and temporal resolution and, in the case of invasive methods, compromise plant health, further constraining their applicability.

Detailed comments:

1. During the transfer printing process, it seems that graphene oxide (GO) was used as the interfacial layer between Ag NWs and aluminum oxide (AAO). After the transfer printing, does GO remain on Ag NWs or on AAO? What is the vacuum pressure used to form the Ag NW film? More experimental data is needed.

Reply:

We thank the reviewer for the very careful review of the manuscript. In our method, to facilitate the release of the vacuum-filtrated AgNW film from the aluminum oxide (AAO) membrane, a small amount of graphene oxide (GO) was introduced as an interfacial layer. This uniform GO layer aids in separating the AgNW film from the AAO membrane, allowing the film to float on the water surface for the subsequent transfer printing. After the transfer printing process, the GO layer remains beneath the AgNW layer.

It is important to note that the amount of GO used is minimal—50 μ L (0.02 wt%) compared to 30 μ L (0.5 wt%) of Ag NWs. This results in an ultrathin GO layer (\sim 8 nm), which is negligible relative to the total thickness of the e-tattoo (\sim 100 nm). Therefore, the e-tattoo demonstrates exceptional impedance measurement capabilities, with performance approximately one to two orders of magnitude lower than that of conventional metal materials (Fig. 4a-f).

The vacuum pressure used during the formation of the Ag NW film was 24 mbar. *To address the reviewer’s comments, we have added further experimental details to the Methods section (page 13), clarifying the role of the GO layer, its influence on the transfer printing process, and the vacuum conditions used.* We hope this additional information provides clarity and addresses the reviewer’s concerns.

“To facilitate the detachment of the AgNW film from the AAO membrane after filtration, a thin layer of graphene oxide (GO) is introduced as an interfacial layer. First, prepare the GO suspension by diluting 50 μ L of a GO dispersion (which is diluted 1:20 with DI water) in 250

mL of pure water. Stir the GO suspension for 1 minute to ensure uniform distribution. Pour the GO suspension into the filtering cup and initiate filtration by turning on the vacuum pump at a pressure of 24 mbar. When 20–30 mL of the GO suspension remains in the filtering cup, carefully add the AgNW suspension and continue the filtration process until all the water has evaporated. The introduced GO on AAO film facilitates the release of the AgNW film from the AAO membrane, allowing the film to float on the water surface for subsequent transfer printing. The GO layer remains beneath the AgNW layer after transfer printing.”

2. In Fig S3, it seems that a mask could be used to create different patterns for the e-tattoo on the leaf. One question is that since the Ag NW film deposited on the mask is continuous, how could the peel-off action of the mask early break up the Ag NW film and leave a pattern behind. If the Ag NW film can be easily broken, does it imply the film might be fragile to mechanical damage?

Reply:

We thank the reviewer for their insightful comments. The patterning method used in our study is a mechanical lift-off process, which is conceptually similar to chemical lift-off techniques commonly employed in micro- and nano-electronics. During the mask removal, localized mechanical stress at the edges of the mask selectively breaks the Ag NW film, enabling the desired patterning. Importantly, this localized breakage during the lift-off process is not indicative of the film’s mechanical fragility under practical usage conditions.

In real-world applications, the Ag NW film operates in a much gentler mechanical environment and is not subject to the concentrated stresses associated with the lift-off procedure. The film exhibits excellent flexibility and adhesion to the leaf surface, ensuring mechanical robustness during normal use.

As demonstrated in Fig. 3b, the Ag NW film showed negligible changes in relative resistance when subjected to repeated bending cycles from 30° to 180°, both upward and downward, confirming its durability and flexibility. Additionally, Fig. 3g highlights the e-tattoo’s ability to conform seamlessly to the leaf’s surface even under extreme deformation, such as dehydration and pronounced wrinkling. Remarkably, the e-tattoo maintained conformal attachment with minimal fluctuations in relative resistance (<5%) during these conditions. These results collectively validate the e-tattoo’s mechanical durability and ensure the reliability of its impedance measurements in practical on-leaf monitoring applications.

3. For the repeatability of leaf impedance measurement, would the location and geometry of the Ag NW electrodes matter? For instance, how far the two electrodes should be spaced on the leaf? Do different plant tissues (young vs. old leaves) affect the baseline impedance value?

Reply:

We thank the reviewer for raising this important point. The location and geometry of the AgNW electrodes indeed influence the impedance values, as the leaf impedance spectrum reflects both the intrinsic properties of the leaf (e.g., tissue composition, ion concentration) and the current pathways established between the electrodes. While the heterogeneity of leaf tissues can induce

slight differences in impedance values, the electrodes geometry and spacing have a more pronounced impact. To address the reviewer's question, we have conducted additional experiments and included the new results in the revised Supplementary Fig. 13,14.

Supplementary Fig. 13. Effect of electrode spacing on leaf impedance measurements. a, Schematic showing different electrode spacing distances ($d = 1.5$ mm, 3 mm, and 5 mm) on leaves. b, Impedance magnitude spectra of leaf measured at different electrode spacings. c, Phase spectra of leaf at different electrode spacings. d, Relative impedance magnitude at 5 kHz of the leaf measured with e-tattoo electrodes at distances of 1.5 mm, 3 mm, and 5 mm, normalized to the impedance at a 1.5 mm distance.

Supplementary Fig. 14. Effect of electrode width on leaf impedance measurements. a, Schematic

and photograph illustrating the arrangement of e-tattoo electrodes; **b**. Impedance magnitude spectra obtained using e-tattoo electrodes with widths of 1 cm, 1.5 cm, and 2 cm. **c**. Phase spectra recorded with e-tattoo electrodes at widths of 1 cm, 1.5 cm, and 2 cm. **d**. Relative impedance magnitude at 5 kHz of the leaf measured with e-tattoo electrodes at widths of 1 cm, 1.5 cm, and 2 cm, normalized to the impedance with a 1 cm width.

As shown in Supplementary Fig. 13, increasing the electrode distance from 1.5 mm to 5 mm leads to higher impedance magnitudes due to the longer current pathway and increased resistance. Similarly, Supplementary Fig. 14 demonstrate that increasing the electrode width from 1 cm to 2 cm decreases impedance magnitudes, due to the larger cross-sectional area available for current flow. Despite these variations, we observed that the relative differences in impedance measurements remained within 30%, indicating that leaf impedance measurements are reasonably tolerant to changes in electrode location and geometry.

We would also like to point out that for monitoring plant immune responses, the primary metric of interest is the relative change in impedance over time rather than the absolute impedance values. This ensures that variability due to leaf heterogeneity and electrodes does not significantly impact the results. Consistent trends in impedance changes before and after immune response activation were observed across three independent batches of *A. thaliana* plants, confirming the robustness of our approach (Supplementary Fig. 23-34).

Regarding the effects of different plant tissues (e.g., young vs. old leaves) on baseline impedance values, we have included a new dataset in Supplementary Fig. 15, which shows EIS measurements for seven leaves of varying sizes and growth stages. The data indicate baseline differences in impedance across leaves, which can be attributed to the natural heterogeneity of leaf tissues (e.g., variations in tissue composition and ion concentration). These differences are normal and well-documented phenomena in plant biology, and do not hinder our ability to monitor physiological changes, such as immune response activation, as the focus remains on tracking relative changes in impedance over time.

Supplementary Fig. 15. EIS of *A. thaliana* leaves with different sizes and growth stages. a, Impedance magnitude spectra obtained from different leaves. **b**, Phase spectra obtained from different leaves.

We have included a paragraph in the Results section (page 8) to clarify the above-mentioned

points:

“The influence of the e-tattoo electrode’s location and geometry on EIS data is presented in Supplementary Fig. 13, 14, demonstrating that leaf impedance measurements are reasonably tolerant to variations in electrode placement and design. Additionally, EIS data for leaves of different sizes and growth stages are provided in Supplementary Fig. 15, while EIS data for different leaves on the same plant are shown in Supplementary Fig. 16. These results highlight baseline differences in impedance across leaves, which can be attributed to the natural heterogeneity of leaf tissues, such as variations in tissue composition and ion concentration.”

4. For the biocompatibility test (Fig 2a), both plants with or without the tattoo patch seem to be under stress. Several leaves turned yellow after 7 days of growth. Any explanation? For the 60-day test, are photos showing the same plant? The container has obviously been changed.

Reply:

We thank the reviewer for the very careful review of the manuscript. The plants shown in Fig. 2a were inducible autoimmune plants, which are known to exhibit a slightly leaky effect even in the absence of an inducing agent. We tested three individual plants with the e-tattoo, and all exhibited phenotypes similar to those of plants without the e-tattoo attached.

As shown in Fig. 2a, both groups—plants with and without the e-tattoo—displayed comparable progression of leaf senescence over seven days. This is particularly evident in the yellowing of older leaves, which is consistent with the plants reaching the flowering stage. During this developmental phase, nutrients are naturally redirected towards the reproductive organs, leading to senescence and chlorosis in older leaves.

Importantly, the observed stress symptoms were not exacerbated in the plants with the e-tattoo compared to those without, suggesting that the e-tattoo did not induce additional stress. The similar patterns of leaf yellowing and overall growth observed in both groups further support this conclusion. We have revised the description in Results section (page 5) to clarify this point:

“The yellowing of older leaves in Fig. 2a reflects natural leaf senescence during the flowering stage, as nutrients are redirected toward reproductive organs, a process observed in both plants with and without the e-tattoo.”

Regarding the 60-day test, we confirm that the photos depict the same plant. The container was swapped out to accommodate the plant’s growth. As shown in the photo, the plant developed more roots over time, necessitating the use of a larger container to support its continued growth and minimize constraints imposed by the original container.

5. In Fig 2d, does the SEM really show a stoma? More SEM images are needed to prove the opening is indeed a stoma.

Reply:

We thank the reviewer for raising this important point. Stomata consist of minute pores, called

stoma, which are surrounded by a pair of guard cells. The structure of stomata has been extensively observed in microscopic images, as shown in below figure.

Fig. R1. Microscopic image of a Pothos leaf's abaxial surface.

In response to the reviewer's suggestion, we have revised Fig. 2d by including an additional SEM image of a pristine stomata to further support our findings. We believe this addition enhances the clarity and validity of the presented data.

Fig. 2d, SEM image of open stomata on the abaxial surface of a pothos leaf. The left panel shows a pristine leaf, while the right panel displays a leaf printed with the e-tattoo. Scale bar, 5 μm.

6. *What is the transparency of the Ag NW film? A UV-vis absorption spectrum of the Ag NW film should be provided.*

Reply:

We thank the reviewer for the insightful suggestion. The transparency of the AgNW film, especially in the visible spectrum, is crucial for non-disruptive on-leaf monitoring. To address this, we have measured the transmission spectrum of the AgNW film in the visible light range, and the results are provided in new Supplementary Fig. 3.

Supplementary Fig. 3. Transmission spectrum of the AgNW film.

As shown in the spectrum, the AgNW film demonstrates a transmittance exceeding 70% across the 400–700 nm wavelength range, which ensures minimal interference with natural light absorption by the leaf. Additionally, we have evaluated the long-term biocompatibility of the Ag NW film when attached to the upper surface of leaves. As demonstrated in Fig. 2f and Supplementary Fig 9, 10, the Ag NW film does not cause any observable tissue damage over a two-week period, unlike films with poor optical transparency.

7. In Fig 2f, where does the fluorescence come from? Is the fluorescence from leaf chlorophyll? What are the excitation and emission wavelengths?

Reply:

We thank the reviewer for raising this important point. The fluorescence shown in Fig. 2f was detected using the ChemiDoc MP Imaging System, which was utilized to capture the autofluorescence emitted by dead or stressed leaf tissue at the indicated time points. The fluorescence does not originate from chlorophyll but instead represents tissue damage or cell death. Representative leaves are shown using a false color scale, where healthy tissue is represented by cooler colors (black to blue) and stressed or dead tissue by warmer colors (orange to white) [R1].

Fig. 2f. Fluorescence image of the *Brassica rapa* leaf after removing the e-tattoo and black tape, showing the e-tattoo-covered area (left) and the black tape-covered area (right).

This imaging approach enables differentiation between healthy and stressed leaf areas, as seen in Fig. 2f, where the stressed regions correspond to higher fluorescence intensity. For the detection, the excitation and emission wavelengths used were specific to the Alexa488 channel, which typically excites around 488 nm and emits around 519 nm. This methodology provides a sensitive means to visualize and quantify stress responses in the leaves over time. We have included a paragraph in the Methods section (page 14) to clarify this point:

“The fluorescence image was detected using the Bio-Rad ChemiDoc MP Imaging System. The fluorescence signal was detected using excitation and emission wavelengths specific to the Alexa Fluor 488 channel, with excitation at approximately 488 nm and emission at approximately 519 nm. For quantification, grayscale fluorescence images were first acquired from the Bio-Rad ChemiDoc MP Imaging System., with pixel intensity values ranging from 0 (black) to 255 (white), corresponding to fluorescence intensity. These grayscale images were then processed using ImageJ, where a false-color scale was applied to map intensity values to colors, enhancing the visual interpretation of the data.”

[R1] Saile, Svenja C., et al. "Two unequally redundant" helper" immune receptor families mediate Arabidopsis thaliana intracellular" sensor" immune receptor functions." *PLoS biology* 18.9 (2020): e3000783.

8. In Fig 4b and 4e, there are signal bumps in several impedance spectra at the frequency of around 10^3 Hz. What is the cause of that?

Reply:

We thank the reviewer for their insightful question. The signal bumps observed around 10^3 Hz in the impedance spectra of Fig. 4b and 4e are primarily due to imperfections in the electrode-tissue interface. These imperfections arise from non-ideal contact between the electrodes and the leaf surface, as the electrodes may not conform perfectly to the leaf's curved and heterogeneous surface. This results in local impedance variations, which is a well-known issue in electrochemical measurements where poor electrode contact can introduce artifacts in the data.

To elaborate, the electrical properties of the electrochemical interface between an electrode and a conductive medium are commonly described by models like the Randles equivalent circuit [R1-R2]. This model represents the interface with a combination of resistance and capacitance, which often results in a characteristic impedance response with a small hump in the middle frequency range (typically between 10^2 Hz and 10^3 Hz). This hump is observed under ideal conditions, where the electrode-tissue interface is uniform and well-defined.

However, as seen in our EIS measurements (Fig. 4b and 4e), imperfections at the electrode-tissue interface lead to deviations from this ideal behavior. Specifically, areas of poor contact increase charge transfer resistance and decrease capacitance, leading to more pronounced signal bumps in the middle frequency range (around 10^3 Hz). These bumps are a direct result of inefficient charge transfer and localized impedance heterogeneity.

Further support for this explanation can be seen when comparing the data in Fig. 4b and Fig.

4e. In Fig. 4e, which shows measurements from *A. thaliana* with trichome-covered (hairy) surfaces, the electrode-tissue contact is even more uneven due to the irregularities and increased surface roughness. This results in more signal bumps, as the contact imperfections are more severe in this case, leading to greater variations in impedance.

In summary, the signal bumps observed around 10^3 Hz are caused by non-uniform electrode-tissue contact, which increases charge transfer resistance and alters capacitive behavior at the electrode-tissue interface. This is a common challenge in bioimpedance measurements, especially when dealing with non-smooth and irregular surfaces such as plant leaves. Our e-tattoo technology, however, forms a superior conformal contact with the leaf tissue, resulting in higher-quality EIS data. This demonstrates the clear advantage of the proposed e-tattoo over conventional electrode-based approaches. We have included a paragraph in the Results section (page 8) to clarify this point:

*“These anomalies are primarily attributed to imperfections at the electrode-tissue interface, where non-uniform contact introduces localized impedance variations. This phenomenon, common in electrochemical measurements, can be explained by deviations from the ideal impedance response modeled by the Randles equivalent circuit^{32,35}. The uneven surfaces of leaves, particularly the trichome-covered *A. thaliana* leaves, exacerbate these imperfections, resulting in more pronounced signal irregularities.”*

[R1] Magar, Hend S., Rabeay YA Hassan, and Ashok Mulchandani. "Electrochemical impedance spectroscopy (EIS): Principles, construction, and biosensing applications." *Sensors* 21.19 (2021): 6578.

[R2] Stupin, Daniil D., et al. "Bioimpedance spectroscopy: basics and applications." *ACS Biomaterials Science & Engineering* 7.6 (2021): 1962-1986.

9. During the immune response of the plant to ethanol exposure, what actually contributed to the decrease in leaf impedance value? Is it related to plant cell death and increased conductivity? More clear explanation of direct factors that lead to leaf impedance decrease is needed.

Reply:

We thank the reviewer for their insightful question. The observed decrease in leaf impedance is primarily driven by the activation of the plant's immune response, particularly through the expression of *RPW8.1^{KZ10}*, which is linked to CNL-mediated hypersensitive response (HR). This activation triggers a cascade of early cellular changes, including ion leakage, membrane disruption, and loss of cell integrity, which are hallmark features of CNL-related HR.

CNLs (Coiled-Coil Nucleotide-Binding Leucine-Rich Repeat receptors) play a crucial role in plant immunity by initiating a strong HR-mediated programmed cell death (PCD). During CNL-activated HR, several physiological and biochemical changes occur, including:

1. Rapid and Localized Cell Death (PCD) – Cellular collapse, nuclear condensation, and breakdown of cellular structures.
2. Ion Leakage and Membrane Disruption – Loss of plasma membrane integrity, resulting in increased ion leakage, which correlates with lower leaf impedance.

3. ROS Burst and Oxidative Stress – Excessive reactive oxygen species (ROS) production amplifies membrane damage and lipid peroxidation.
4. Calcium Signaling and MAPK Activation – Calcium influx activates MAPK cascades (MPK3/MPK6), leading to transcriptional defense responses.
5. Early Activation of Defense Genes – Upregulation of salicylic acid (SA) signaling, WRKY transcription factors, and PR genes.
6. Metabolic Reprogramming – Changes in primary metabolism and accumulation of secondary metabolites (phytoalexins, flavonoids).

These HR-associated changes, particularly membrane disruption, increased ion conductivity, and PCD, collectively drive the decrease in leaf impedance observed in our study. To support this interpretation, Fig. 5d shows a significant increase in conductivity (indicative of ion leakage) within 3 hours, which directly correlates with a decrease in leaf impedance. This confirms the relationship between CNL-mediated cell death, ion flux changes, and impedance reduction. Previous studies have also demonstrated that membrane integrity loss significantly contributes to impedance shifts, further reinforcing our findings [R1].

To clarify this point, we have added a detailed explanation in the Results section (page 10) of the revised manuscript:

“The observed decrease in leaf impedance during the plant's immune response to ethanol exposure is primarily driven by cellular changes associated with the activation of RPW8.1^{KZ10}, which activates a CNL-related immune receptor in the transgenic line. This activation triggers a hypersensitive response (HR), a well-characterized form of programmed cell death. HR involves early physiological changes, including ion leakage, loss of cell membrane integrity, and reactive oxygen species (ROS) accumulation, which collectively contribute to the decrease in leaf impedance.”

[R1] Stupin, Daniil D., et al. "Bioimpedance spectroscopy: basics and applications." *ACS Biomaterials Science & Engineering* 7.6 (2021): 1962-1986.

10. What is the voltage of the alternating electrical field?

Reply:

We thank the reviewer for raising this point. The voltage of the alternating electrical field is set to 0.7 V. This information has been included in the Methods section (page 14) of the revised manuscript.

“The full impedance spectrum was measured across a frequency range of 1 Hz to 1 MHz using an electrochemical workstation (CHI600E). The voltage of the alternating electrical field is set to 0.7 V.”

11. The conclusion paragraph mentioned that the sensor could detect plant biological response within ~3 hours, while the abstract mentioned it was within 5 hours. This inconsistency needs

to be corrected.

Reply:

We thank the reviewer for pointing out this inconsistency. This value has been corrected in the revised manuscript. We have carefully gone over the manuscript to ensure that all the values are correct.

“We have successfully demonstrated continuous EIS analysis of live transgenic Arabidopsis thaliana plants for over 24 hours, capturing the onset of NLR-mediated acute immune responses within three hours post-induction, prior to visible symptoms.”

Reviewer #2 (Remarks to the Author):

Reply:

We thank the reviewer for the careful review of the manuscript and constructive feedback. We believe that the manuscript is much improved as a result.

Reviewer #3 (Remarks to the Author):

In this manuscript, He et al. described a silver nanowire-based, substrate-free electrode for monitoring the health of plants using impedance spectroscopy. The authors demonstrated the attachment of the e-tattoo onto the leaves of Arabidopsis to measure their electrochemical impedance under different immune conditions. While continuous, non-invasive monitoring of plant health is quite interesting, there are several major issues that require careful attention:

Reply:

We thank the reviewer for noting the general interest of our work for continues, non-invasive plant health monitoring.

1. The manuscript lacks a clear explanation of the reasoning behind impedance changes in plant leaves. From the images presented in Figure 1, it seems that the reference electrode (REF) and working electrode (WE) are both on the same leaf and adjacent to each other. The observed impedance changes could simply reflect the hydration state of the leaf, which might correlate with the immune response. What is the main mechanism driving these impedance changes?

Reply:

We thank the reviewer for raising this important point. Unlike animal immune systems, which rely on specialized organs or localized tissues, plant immune responses are systemic, involving activation throughout the entire organism. In our study, plants were engineered to trigger a systemic immune response upon ethanol induction. As a result, measuring immune responses in a single representative leaf is a widely accepted practice for assessing plant-wide immune activation.

The observed decrease in leaf impedance is primarily driven by the activation of the plant's immune response, particularly through the expression of *RPW8.I^{KZ10}*. This activation leads to early cellular changes, such as ion leakage and membrane disruption, which are characteristic features of the hypersensitive response (HR). HR is a well-established form of programmed cell death, marked by rapid cell death, disruption of ion balance, and the breakdown of cellular components. These physiological changes directly influence leaf impedance, which reflects a combination of intrinsic properties such as tissue composition, ion concentration, and membrane integrity.

In Fig. 5d, we quantified cell death triggered by *RPW8.I^{KZ10}*. The results show a significant increase in conductivity (indicating higher ion leakage) within 3 hours, which directly correlates with a decrease in leaf impedance. This increase in conductivity is linked to the loss of cell membrane integrity, a hallmark of HR. The temporal relationship between ion leakage and impedance reduction suggests a direct connection between cellular death and the physiological changes induced by ethanol exposure. Additionally, prior studies have shown that cell membrane disruption significantly contributes to impedance reduction, consistent with our findings [R1]. To clarify this point, we have added a detailed explanation in the Results section (page 10) of the revised manuscript:

“The observed decrease in leaf impedance during the plant's immune response to ethanol exposure is primarily driven by cellular changes associated with the activation of RPW8.1^{KZ10}, which activates a CNL-related immune receptor in the transgenic line. This activation triggers a hypersensitive response (HR), a well-characterized form of programmed cell death. HR involves early physiological changes, including ion leakage, loss of cell membrane integrity, and ROS accumulation, which collectively contribute to the decrease in leaf impedance.”

To rule out the possibility that the observed impedance changes were due to variations in leaf hydration, we have performed additional experiments measuring the electrical impedance spectroscopy (EIS) of *A. thaliana* leaves at different dehydration levels. We would like to point out that prior to immune activation, the plants were fully hydrated, ensuring consistent and optimal hydration across all leaves. If hydration were responsible for the impedance change, we would expect a clear correlation with water content loss. However, when leaves lost approximately 44% of their water content (LWC), the impedance increased by only 11.9%, a much smaller change than the significant reduction observed with immune response activation. Moreover, dehydration resulted in an impedance increase, rather than a decrease of immune response. These results have been summarized in a new supplementary Fig. 21 and strongly indicate that the observed impedance reduction is not due to hydration changes but instead arises directly from HR-induced cellular death and associated physiological changes.

Supplementary Fig. 21. EIS response of plant leaves to dehydration. **a**, Impedance magnitude spectra of a *A. thaliana* leaf at different levels of LWC (loss of water content). **b**, Phase spectra of a *A. thaliana* leaf at different levels of LWC.

To clarify this point, we have added a detailed explanation in the Results section (page 9-10) of the revised manuscript:

*“Additionally, we tested the impedance changes in response to other common stressors, such as dehydration and wounding, to eliminate the possibility of these factors contributing to the observed changes (Supplementary Fig. 21, 22). Notably, water loss and wounding induced only minor upward shifts in the impedance magnitude spectrum, with small and distinct variation trends. These findings further validate that the observed impedance changes in the transgenic *A. thaliana* are specifically induced by the immune response rather than other factors.”*

[R1] Stupin, Daniil D., et al. "Bioimpedance spectroscopy: basics and applications." *ACS Biomaterials Science & Engineering* 7.6 (2021): 1962-1986.

2. Although the authors achieved conformable contact with the leaf, it remains unclear how a long-term, reliable ohmic contact is maintained with the e-tattoo itself. Demonstrating continuous longitudinal measurements would be crucial to highlight the advantages of the e-tattoo.

Reply:

We appreciate the reviewer's insightful comment and suggestion. We agree that maintaining long-term, reliable ohmic contact is crucial for the practical application of the e-tattoo, and this is one of the key focuses of our study. The e-tattoo's ultra-thin design (~100 nm) and low sheet resistance (5 Ω /square) enable excellent conformal contact with the leaf tissue, as demonstrated in Fig. 2c-d. This conformability ensures high-quality ohmic contact, which is critical for acquiring accurate EIS spectra, as further supported by comparisons with conventional electrodes in Fig. 4a-f.

To assess the long-term stability of the e-tattoo, we performed EIS measurements on intact plants over a 14-day period (Fig. 4i, j, and Supplementary Fig. 17,18). The results showed consistent EIS data across three separate trials, confirming that the e-tattoo maintains reliable ohmic contact with the leaf tissue over an extended period. Additionally, Fig. 5b demonstrates continuous longitudinal impedance measurements, tracking impedance changes during immune response activation.

In response to the reviewer's suggestion, we have included a new Supplementary Fig. 20 that provides additional data on continuous longitudinal impedance measurements for 5 days. This new dataset further illustrates the feasibility and robustness of the e-tattoo for long-term monitoring, with the measurement duration adaptable to specific experimental needs. We have also included the corresponding description in the Results section on page 8, as provided below.

Supplementary Fig. 20. Continuous impedance monitoring of two leaves on intact *A. thaliana* plants for 5 days.

“Furthermore, continuous impedance measurements over a 5-day period further demonstrate the robustness and feasibility of the e-tattoo for long-term monitoring, with measurement durations adaptable to specific experimental needs (Supplementary Fig. 20).”

3. *The manuscript lacks sufficient statistical analysis. The group size, metrics for comparison, and the significance of the biocompatibility tests are unclear. Is biocompatibility solely assessed through images of leaves with and without the e-tattoo?*

Reply:

We thank the reviewer for raising this point. Ensuring the biocompatibility of plant wearable sensors is critical for their successful integration with plant tissues, especially for long-term physiological monitoring.

Visual observation is a widely accepted method for assessing biocompatibility in plant wearable sensor studies [R1-R4], as it allows for the detection of clear signs of stress or adverse reactions, such as leaf discoloration, wilting, necrosis, or changes in growth patterns. This method is non-invasive, quick, repeatable, and does not require destructive sampling, enabling real-time evaluation of plant health and sensor performance under natural growth conditions.

However, we would like to emphasize that visual assessment was not the sole method used to evaluate biocompatibility in our study. In addition to visual observations, we employed fluorescence imaging of plant leaves to assess tissue viability. This technique detects autofluorescence emitted by stressed or damaged tissues, providing direct evidence of biocompatibility. The fluorescence imaging results, summarized in the revised Fig. 2, confirm that the e-tattoo does not induce significant stress or damage to the plant tissue. We have added clarifications regarding these methods in the Methods section of the revised manuscript:

“The fluorescence image was detected using the Bio-Rad ChemiDoc MP Imaging System. The fluorescence signal was detected using excitation and emission wavelengths specific to the Alexa Fluor 488 channel, with excitation at approximately 488 nm and emission at approximately 519 nm. For quantification, grayscale fluorescence images were first acquired from the Bio-Rad ChemiDoc MP Imaging System., with pixel intensity values ranging from 0 (black) to 255 (white), corresponding to fluorescence intensity. These grayscale images were then processed using ImageJ, where a false-color scale was applied to map intensity values to colors, enhancing the visual interpretation of the data.”

Fig. 2f. Fluorescence image of the *Brassica rapa* leaf after removing the e-tattoo and black tape, showing the e-tattoo-covered area (left) and the black tape-covered area (right).

We have also included new Supplementary Fig. 10 to illustrate the quantified fluorescence intensity variations between the e-tattoo-covered area and the tape-covered area.

Supplementary Fig. 10. Comparison of the impact of the e-tattoo and black tape on leaf tissue health. Brightness values of individual pixels were quantified by adjusting the intensity range of the entire image to 55–255. The brightness intensities of the areas covered by the e-tattoo and black tape were analyzed for **a**, *Brassica rapa* leaf, and **b**, Pothos leaf. Error bars show mean \pm s.d. (n = 10 samples).

Furthermore, to validate the long-term biocompatibility of the e-tattoo, we conducted continuous on-plant EIS measurements on *A. thaliana* over a two-week period (Fig. 4k-1, Supplementary Fig. 17-19). The results showed consistent impedance values throughout the monitoring period, indicating stable physiological conditions and demonstrating that the e-tattoo maintains reliable, long-term contact with the leaf without causing significant physiological changes. These assessments also validate the biocompatibility of the e-tattoo and its suitability for non-invasive, long-term monitoring of plant physiological states.

[R1] Kim, Jae Joon, Linden K. Allison, and Trisha L. Andrew. "Vapor-printed polymer electrodes for long-term, on-demand health monitoring." *Science advances* 5.3 (2019): eaaw0463.

[R2] Kim, Jae Joon, et al. "On-site identification of ozone damage in fruiting plants using vapor-deposited conducting polymer tattoos." *Science Advances* 6.36 (2020): eabc3296.

[R3] Chai, Yangfan, et al. "Cohabiting plant-wearable sensor in situ monitors water transport in plant." *Advanced Science* 8.10 (2021): 2003642.

[R4] Yang, Yanqin, et al. "All-organic transparent plant e-skin for noninvasive phenotyping." *Science Advances* 10.7 (2024): eadk7488.

4. *The impedance measurements are performed on only one leaf. Would this single-leaf approach be representative of the overall plant health? Additionally, what is the variability in impedance across different leaves of the same plant?*

Reply:

We appreciate the reviewer's insightful comment. In plants, immunity is not confined to specialized organs or tissues, distinguishing it from typical animal immunity. In our study, the plants were engineered to induce immune responses throughout the entire organism upon ethanol exposure. Therefore, measuring immune responses in a single representative leaf is a

well-established and accepted approach.

In *A. thaliana* research, it is standard to sample leaves at specific developmental stages, such as L5 or L6, which are sequentially positioned from the shoot apical meristem. Leaves L5 to L7 are particularly suitable for such measurements as they develop a characteristic flat, laminar structure with distinct vein patterns. Their epidermal cells differentiate into trichomes (hair-like structures), stomata, and pavement cells. These leaves, at an intermediate stage of maturity, balance active growth with well-established defense mechanisms and are pivotal in studying plant immunity in *A. thaliana* [R1]. Since *A. thaliana* typically has 10-15 leaves, selecting a single leaf (representing ~10% of the total leaf area) provides a sufficiently representative measurement of overall plant health and immune status.

To address the reviewer's concern about variability in impedance across different leaves on the same plant, we have included a new Supplementary Fig. 16 showing that impedance differences between leaves on the same plant is less than 20%. This variability arises from natural heterogeneity in leaf tissues, such as differences in ion concentration and tissue composition, which is well-documented in plant biology. However, as previously noted, these baseline variations do not affect our ability to track real-time physiological changes. Our analysis focuses on relative impedance changes over time, which allows for accurate monitoring of immune responses, regardless of the baseline differences between leaves.

Supplementary Fig. 16. EIS of different leaves on the same *A. thaliana* plant. a, Impedance magnitude spectra obtained from different leaves. **b,** Phase spectra obtained from different leaves.

[R1] Li, Lei, et al. "Protein degradation rate in Arabidopsis thaliana leaf growth and development." *The plant cell* 29.2 (2017): 207-228.

5. How did the authors quantify the color bar in Figure 2F? It is unclear how the fluorescence image was colored and at what wavelength it was captured.

Reply:

We thank the reviewer for raising this point for clarification. The fluorescence shown in Fig. 2f was detected using the Bio-Rad ChemiDoc MP Imaging System, which captures the autofluorescence emitted by stressed or dead leaf tissue at the indicated time points. The representative leaves in the figure are displayed using a false-color scale, where cooler colors

(black to blue) indicate healthy tissue, and warmer colors (orange to white) represent stressed or dead tissue [R1].

The fluorescence signal was detected using excitation and emission wavelengths specific to the Alexa Fluor 488 channel, with excitation at approximately 488 nm and emission at approximately 519 nm. For quantification, grayscale fluorescence images were first acquired from the Bio-Rad Alexa Fluor 488 channel, with pixel intensity values ranging from 0 (black) to 255 (white), corresponding to fluorescence intensity. These grayscale images were then processed using ImageJ, where a false-color scale was applied to map intensity values to colors, enhancing the visual interpretation of the data. To provide further clarity, we have included a new Supplementary Fig. 10, which illustrates the comparison of intensity values.

This methodology provides a sensitive means to visualize and quantify stress responses in the leaves over time. We have included a paragraph in the Methods section to clarify this point:

“The fluorescence image was detected using the Bio-Rad ChemiDoc MP Imaging System. The fluorescence signal was detected using excitation and emission wavelengths specific to the Alexa Fluor 488 channel, with excitation at approximately 488 nm and emission at approximately 519 nm. For quantification, grayscale fluorescence images were first acquired from the Bio-Rad ChemiDoc MP Imaging System., with pixel intensity values ranging from 0 (black) to 255 (white), corresponding to fluorescence intensity. These grayscale images were then processed using ImageJ, where a false-color scale was applied to map intensity values to colors, enhancing the visual interpretation of the data.”

[R1] Saile, Svenja C., et al. "Two unequally redundant" helper" immune receptor families mediate Arabidopsis thaliana intracellular" sensor" immune receptor functions." *PLoS biology* 18.9 (2020): e3000783.

6. *A major advantage of substrate-free electrodes is their potential to stretch and grow with the leaf. Given the capability of the existing electrodes to perform electrochemical impedance spectroscopy (EIS), it is critical to demonstrate the added value of these electrodes. Their stability and measurement capability at different stages of growth should be validated. Furthermore, is a 24-hour timeline physiologically relevant for such plants?*

Reply:

We thank the reviewer for the valuable feedback. We agree that substrate-free electrodes have the potential to stretch and grow with the leaf, which is a notable advantage. However, we would like to point out that the primary strengths of our proposed e-tattoo lie in its ultrathin design, excellent conformability, biocompatibility, and high conductivity. These features collectively enable non-invasive, high-quality EIS data collection with long-term monitoring capability.

To address the reviewer’s concern, we have also added a new Supplementary Fig.19, demonstrating the long-term EIS measurements of *A. thaliana* plants (over three weeks). The data show that the impedance spectra maintain a consistent shape, with only a slight increase

in impedance values, likely due to the leaf's continuous growth and development. These results demonstrate the stability and reliability of the e-tattoo for monitoring leaf impedance over extended periods.

Supplementary Fig. 19. EIS data collected on two intact *A. thaliana* plants over a 21-day period. **a**, Impedance magnitude spectra in different days of a *A. thaliana* plant. **a**, Impedance magnitude spectra in different days of another *A. thaliana* plant.

Regarding the 24-hour timeline, we consider it physiologically relevant. RNA-seq data from our study indicate that the immune response in *A. thaliana* begins as early as 3 hours post-induction (hpi). This early response is reflected in the observed increase in conductivity, which is evident at around the same time. Thus, a 24-hour measurement period is suitable, as it allows for the full progression of immune signaling, including physiological changes such as cell death, ion leakage, and tissue remodeling. This timeline effectively captures both the onset and resolution of the immune response, providing a comprehensive understanding of the physiological processes involved. Besides, as plants exhibit diurnal rhythms in their physiological activities. Monitoring over this timeframe helps ensure that natural variations due to these rhythms do not confound the impedance measurements. We have revised the description in the Results section (page 9) to clarify this point:

“The data collected around 24 hours post-induction captures the full progression of immune signaling, encompassing key physiological changes such as cell death, ion leakage, and tissue remodeling. This timeline effectively reflects both the onset and resolution of the immune response, offering a comprehensive view of the underlying physiological processes.”

Reviewer #4 (Remarks to the Author):

In the manuscript entitled “Epidermal electronic-tattoo for plant immune response monitoring,” He et al. report an ultrathin, substrate-free, and highly conductive electronic tattoo (e-tattoo) designed for plants, enabling immune response monitoring through non-invasive and robust electrical impedance spectroscopy (EIS) analysis. The authors demonstrate continuous EIS analysis of live transgenic plants for over 24 hours, capturing the onset of NLR-mediated acute immune responses within five hours post-induction, even before visible symptoms appear. This represents a promising technical advancement with great potential to enhance our understanding of plant immune activation mechanisms.

Reply:

We thank the review for the positive comments regarding the potential applications.

However, the manuscript currently lacks evidence to confirm whether the e-tattoo specifically detects NLR-mediated immune responses. Furthermore, for its broader application in plant immunity research, it is essential to show whether the e-tattoo can monitor physiological and molecular changes in various plants upon exposure to living virulent and avirulent pathogens. Without this validation, the scope of the e-tattoo’s application would be quite limited. Additionally, the manuscript should discuss how this technology could be integrated with advanced molecular and genetic approaches to advance plant immunity research. Addressing these points would significantly strengthen the impact of this invention.

Reply:

We thank the review for the insightful comments. Regarding the first two comments, we note that they overlap with those raised in the subsequent major points, and we will provide detailed responses in the corresponding section of our reply.

To address the reviewer’s comment on integrating this technology with advanced molecular and genetic approaches, we have included a new paragraph in the Results section of revised manuscript on page 11-12. Our plant e-tattoo has demonstrated the capability of capture dynamic immune response-related physiological changes, providing a real-time, continuous, and non-invasive phenotyping tool that complements molecular and genetic approaches. The integration of plant e-tattoo and genetic approaches offers a novel platform for monitoring immune responses at both physiological and biophysical levels, and enables deeper insights into the molecular basis of plant immunity.

*“By integrating RNA-seq analysis with EIS measurements, our study establishes a direct correlation between impedance changes and gene expression dynamics during immune responses. Specifically, our time-course RNA-seq analysis of inducible DM6-DM7 in *A. thaliana* reveals that the rapid induction of RPW8.1^{KZ10}-mVenus, detected as early as 1.5 hpi and peaking at 3 hpi, coincides with initial impedance variations. This suggests that early molecular and cellular changes triggered by RPW8.1^{KZ10} activation directly influence the plant’s electrical properties. Furthermore, the sequential upregulation of EDS16 and PRI aligns with sustained physiological modifications and immune responses, likely contributing*

to continuous impedance shifts. These findings demonstrate that impedance spectroscopy can capture dynamic immune-related physiological changes, providing a real-time, continuous, and non-invasive phenotyping tool that complements molecular and genetic approaches. The integration of plant e-tattoo and genetic approaches offers a novel platform for monitoring immune responses at both physiological and biophysical levels, and enables deeper insights into the molecular basis of plant immunity.”

In addition, we have incorporated new experimental data demonstrating the application of the plant e-tattoo for monitoring another type of NLR-mediated immune response, specifically TNL (Toll/interleukin-1 receptor-like NLR)-mediated immunity, with detailed discussion and results presented in Supplementary Fig. 37-39 and Supplementary Note 2. While both CNLs and TNLs contribute to robust immune signaling, they exhibit distinct structural and functional characteristics. We show that the selected TNL- and CNL-mediated immune responses exhibit distinct kinetic profiles, which can be effectively captured using our e-tattoo technology. CNL-triggered cell death is generally considered to be under direct execution, leading to a rapid hypersensitive response. This notion is reflected in a sharp decrease in impedance magnitude within approximately 3 hours post-induction of the representative CNL-activation system that we established using stable transgenic line carrying *DM6-DM7*. In contrast, TNL-triggered cell death requires a signaling adaptor and small molecules as a second message [R1], and thus is expected to be more prolonged. Our impedance data reveal a distinct pattern for TNL responses: a gradual increase in impedance magnitude, peaking 4-6 days post-induction, followed by a sharp decline that corresponds to severe cell death. This delayed response aligns well with the slower propagation of TNL-mediated immunity compared to CNL.

To further validate the specificity of our e-tattoo in detecting distinct NLR-mediated immune responses, we performed comparative transcriptome analysis between CNL and TNL-activated immunity. Our WGCNA analysis identified five major co-expression modules (ME1-ME5) that exhibited distinct temporal patterns between CNL and TNL activation (Supplementary Fig. 39a). Notably, ME3 and ME5, which represent major immune response signatures, showed contrasting dynamics between CNL and TNL activation (Supplementary Fig. 39b). ME3, enriched in early immune response genes, showed rapid induction in CNL-mediated immunity within 3-6 hours, while displaying delayed activation in TNL-mediated responses. Conversely, ME5 exhibited sustained suppression in CNL responses but gradual recovery in TNL responses. These distinct module patterns were further reflected in the expression profiles of key immune marker genes (Supplementary Fig. 39c). CNL activation triggered rapid induction of defense-related genes, with expression peaks occurring within 3-6 hours post-induction, while TNL-mediated responses showed delayed but sustained activation peaking between 12-24 hours. These distinct transcriptional kinetics align with our impedance measurements, demonstrating that our e-tattoo can effectively distinguish between different NLR-mediated immune responses, thereby validating its specificity and utility for monitoring distinct immune activation patterns in plants.

The integration between electrical phenotyping (via e-tattoo) and genetic analysis could open up new possibilities for studying immune response dynamics both at physiological and biophysical levels, complementing traditional molecular and genetic techniques.

[R1] Qu, ChunChun, et al. "Liquid metal-based plant electronic tattoos for in-situ monitoring of plant physiology." *Science China technological sciences* 66.6 (2023): 1617-1628.

Supplementary Fig. 37. Monitoring TNL-mediated immune response in transgenic *A. thaliana* (Cdm-0) using a plant e-tattoo. **a**, Images of *A. thaliana* before and after ethanol induction to activate the DM10^{TueScha-9} TNL^{L3} in the Cdm-0 background. Four-week-old transgenic Cdm-0 plants carrying pAlcA::DM10^{TueScha-9} were treated with 1% ethanol via water irrigation for seven days under a covered dome. Tissue necrosis became visible after 4 days post-induction (dpi), with the lower image showing necrosis progression at 4 dpi. **b**, Normalized impedance magnitude at 2 kHz over time for the test and control groups following induction with ethanol or water.

Supplementary Fig. 38. Impedance signature of CNL- and TNL-mediated immune response. **a**, Normalized impedance magnitude at 2 kHz of a typical *A. thaliana* (Cdm-0 T₃ pAlcA::DM10^{TueScha-9}) post induction with ethanol. **b-c**, The corresponding bode plots of the control and test plants collected at time points indicated with black and red triangular in **a**. **d**, Normalized impedance magnitude at 2

kHz of a typical *A. thaliana* (Mrk-0 T₃ pAlcA::RPW8.1^{KZ10}) post induction with ethanol. e-f, The corresponding bode plots of the control and test plants collected at time points indicated with black and red triangular in d.

Supplementary Fig. 39. Comparative analysis of CNL and TNL-mediated transcriptional responses. a, Heatmap showing differential gene expression patterns (log₂FC EtOH vs Mock) in CNL (pAlcA::RPW8.1^{KZ10}-mVenus, DM6-DM7) and TNL (pAlcA::DM10^{TueScha-9}, DM10-DM11) immune responses across indicated timepoints. Five major co-expression modules (ME1-ME5) were identified

through WGCNA analysis, with the number of genes in each module indicated. **b**, Temporal dynamics of ME3 (top) and ME5 (bottom) module eigengenes in CNL (pAlcA::RPW8.1^{KZ10}-mVenus, solid lines) and TNL (pAlcA::DM10^{TueScha-9}, dashed lines) immune responses. Shaded areas represent 95% confidence intervals. Blue and red lines represent mock and EtOH treatment, respectively. **c**, Expression profiles of immune marker genes during CNL (pAlcA::RPW8.1^{KZ10}-mVenus, solid lines) and TNL (pAlcA::DM10^{TueScha-9}, dashed lines) activation. Selected genes include ADR1 family members (AT1G33560, AT4G33300, AT5G04720), NRG1.1 (AT5G66900), EDS16 (AT1G74710), and PR1 (AT2G14610). Data points represent mean \pm SEM. Blue and red lines indicate mock and EtOH treatment, respectively.

Major Points

1. *Please provide evidence that the e-tattoo can specifically detect NLR-mediated immune responses. Experiments involving MAMP or DAMP treatments and their effect on EIS analysis would help clarify this point.*

Reply:

We thank the review for raising this important point. We address aspect of the comments in detail:

- **Evidence for NLR-mediated immune response.** The DM6-DM7 NLR pair is a well-established genetic model for studying NLR-mediated autoimmunity. This autoimmunity occurs naturally in hybrid Arabidopsis accessions, where incompatible genetic interactions between DM6 and DM7 lead to spontaneous immune activation. QTL analysis has confirmed that the genomic regions encoding DM6 and DM7 are associated with hybrid necrosis in natural crosses, such as Lerik1-3 \times Fei-0 and Mrk-0 \times KZ10 [R1]. Mechanistic studies have demonstrated that DM7, a homolog of RPW8 (HR4^{Fei-0}), promotes the formation of a higher-order immune receptor complex involving RPP7^{Lerik1-3} (DM6), a coiled-coil NLR (CNL). This complex assembly triggers downstream immune signaling, leading to cell death [R2]. While DM7 alone can induce a weak hypersensitive response (HR)—a form of programmed cell death associated with plant immunity—the simultaneous presence of both DM6 and DM7 results in a significantly stronger HR in both *A. thaliana* and *Nicotiana benthamiana* [R2].

To validate that the observed autoimmunity is specifically mediated by NLR activity, we conducted a genetic experiment. We induced the expression of DM7 RPW8.1^{KZ10} in two distinct backgrounds: (1) Mrk-0, which carries the incompatible RPP7Mrk-0 partner, and (2) Col-0, which lacks this NLR partner. Notably, cell death was observed only in the Mrk-0 transgenic lines, demonstrating that the presence of RPP7 is essential for triggering autoimmunity. This finding provides strong evidence that the immune response we detected is driven by an NLR-specific signaling mechanism rather than non-specific stress responses or other immune pathways.

We have included additional experimental results in new Supplementary Fig. 35 to further

support this conclusion. Phenotypic observations revealed that cell death was visible only in Mrk-0 transgenic lines following induction, while Col-0 transgenic lines remained unaffected (Supplementary Fig. 35a). Western blot analysis confirmed the successful induction of DM7 expression in both genetic backgrounds, as evidenced by the presence of GFP-tagged RPW8.1^{KZ10} (Supplementary Fig. 35b). Despite comparable expression levels, only the Mrk-0 transgenic lines exhibited a robust hypersensitive response, reinforcing the necessity of RPP7 for a strong immune reaction. Additionally, ion conductivity measurements provided quantitative evidence of cell membrane damage, a hallmark of cell death. Conductivity levels increased rapidly and significantly in the Mrk-0 transgenic lines, reflecting severe membrane disruption. In contrast, Col-0 displayed a slower and milder increase in conductivity, consistent with the weaker response expected in the absence of RPP7 (Supplementary Fig. 35c). These observations align with previous findings reported by Li et al. in *Cell Host & Microbe* [R2], further validating the role of the DM6-DM7 pair functions as an NLR-based immune signaling module responsible for triggering autoimmunity.

- **DAMP's effect on EIS analysis.** To assess the impact of DAMPs on EIS signals, we have included new experiments of EIS on wounded tissues, which are a well-known source of DAMPs such as ATP, oligogalacturonides (OGs), and peptides like AtPep1. While direct impedance data from purified DAMP treatments are not yet available, extensive literature evidence indicates the similarity in immune responses triggered by wounding and DAMPs. OGs, derived from the partial hydrolysis of homogalacturonan in the cell wall during wounding, act as key DAMPs, triggering immune responses such as ROS production, MAPK activation, and defense gene expression. These responses are highly analogous to those triggered by PAMPs, indicating significant overlap between DAMP- and PAMP-induced signaling pathways [R3, R4]. The measured data have been summarized in new Supplementary Fig. 22. The results illustrate that although wounding results in a slightly increase of impedance (~0.93% at 5 kHz), this is negligible compared to the significant decrease observed during the NLR-triggered immune activation.
- **Differentiation from other stressors.** To further clarify the specificity of the e-tattoo-enabled EIS measurements, we conducted additional experiments measuring impedance changes due to other stressors, such as wounding and dehydration. These results, detailed in response to **Major Point 2**, confirm that the impedance shifts induced by immune responses are distinct in magnitude and trend from those caused by other stressors.
- **Demonstration of differentiating immune responses.** To illustrate the capability of our e-tattoo in distinguishing different immune responses, we have included new experiments monitoring EIS changes in *A. thaliana* with another induced immune response, triggered by a TNL activation (Supplementary Fig. 37-39 and Note 2). These results are detailed in response to **Major Point 3**. The impedance variation trends for CNL- and TNL-mediated immune responses displayed distinct characteristics in magnitude, direction, and rate of variation. These results correlate well with the transcriptome kinetics of their respective immune signaling pathways, demonstrating that plant e-tattoo can serve as a robust, non-invasive, and continuous monitoring tool for diverse immune responses.

- **Future prospects.** The e-tattoo has the potential to specifically detect additional immune responses, however, further exploration of other immune responses and stressors is beyond the scope of this study. The key novelty of this work lies in the device design, developing an ultrathin, substrate-free, and highly conductive e-tattoo specifically engineered for plants. This device enables long-term, non-invasive EIS monitoring without damaging delicate plant tissues. This work serves as a proof of concept, laying the groundwork for future studies that could expand the analysis to more stressors and immune responses. We thank the reviewer for highlighting this direction and have expanded the Discussion section to address these future prospects.

“While our current study provides a rigorous and controlled framework for characterizing NLR-specific impedance signatures, this work serves as a proof of concept, laying the foundation for future studies applicable to a broader range of immune responses and stressors.”

Supplementary Fig.35. Ethanol-inducible RPW8.1^{KZ10}-GFP triggers cell death in Mrk-0 with paired NLR RPP7^{Mrk-0}. **a**, Phenotypes of T₂ transgenic Arabidopsis populations with ethanol-inducible RPW8.1^{KZ10}-GFP in Col-0 and Mrk-0 genetic backgrounds, observed 7 days post-induction

with 1% ethanol. Photographs were taken of 30-day-old plants. Scale bar: 1 cm. **b**, Western blot analysis of total protein extracted from Arabidopsis leaves collected at the indicated time points (3 to 25 hours post-induction). The expression levels of GFP-tagged RPW8.1^{KZ10} were detected, and possible cleaved or degraded protein bands are marked with asterisks. Ponceau staining confirms comparable protein loading across samples. **c**, Ion leakage measurements as an indicator of cell death severity in the ethanol-inducible Mrk-0 and Col-0 transgenic two lines shown panel **a**.

Supplementary Fig. 22. EIS response of plant leaves to wounding. The wound was created by making a small hole with a tweezer at the midpoint between the two e-tattoo electrodes. EIS measurements were taken 10 minutes post-wounding to assess the response. **a-b**, Impedance magnitude and phase spectra of a sweet potato leaf before and after wounding. **c-d**, Impedance magnitude and phase spectra of another sweet potato leaf before and after wounding with two cuts.

[R1] Barragan, Cristina A., et al. "RPW8/HR repeats control NLR activation in Arabidopsis thaliana." *PLoS genetics* 15.7 (2019): e1008313.

[R2] Li, Lei, et al. "Atypical resistance protein RPW8/HR triggers oligomerization of the NLR immune receptor RPP7 and autoimmunity." *Cell host & microbe* 27.3 (2020): 405-417.

[R3] Savatin, Daniel V., et al. "Wounding in the plant tissue: the defense of a dangerous passage." *Frontiers in plant science* 5 (2014): 470.

[R4] Vega-Muñoz, Isaac, et al. "Breaking bad news: dynamic molecular mechanisms of wound response in plants." *Frontiers in plant science* 11 (2020): 610445.

2. *Testing the response to abiotic stresses, such as wounding or dehydration, would also demonstrate the specificity and robustness of the technology.*

Reply:

We thank the reviewer for their insightful suggestion. In response, we have conducted

additional EIS measurements to evaluate the response of leaves to abiotic stresses, such as wounding and dehydration. The results have been included in the revised manuscript as new Supplementary Fig. 21, 22.

Supplementary Fig. 21. EIS response of plant leaves to dehydration. a, Impedance magnitude spectra of a *A. thaliana* leaf at different levels of LWC (loss of water content). **b,** Phase spectra of a *A. thaliana* leaf at different levels of LWC.

For wounding, we observed a minimal change in impedance ($\sim 0.22\%$ relative change at 5 kHz), which is significantly smaller than the variation induced by immune responses. Additionally, the trend of impedance changes differs: during the immune response, impedance decreases, whereas it increases in response to wounding. Although two cuts result in a slightly larger impedance change ($\sim 0.93\%$ relative change at 5 kHz), this is still much smaller than the variation observed during immune activation.

Regarding dehydration, even when the leaf loses nearly half of its water content (44% LWC), the impedance increases by only 11.9%. This change is not only smaller than that caused by the immune response, but the trend is also reversed.

These findings demonstrate that the EIS data can effectively distinguish between NLR-mediated immune responses and common stress conditions such as wounding and dehydration. This specificity and robustness, as suggested by the reviewer, further validate the utility of the proposed technique. We have incorporated additional details on this point in the revised manuscript's Results section (pages 10-11).

“Additionally, we tested the impedance changes in response to other common stressors, such as dehydration and wounding, to eliminate the possibility of these factors contributing to the observed changes (Supplementary Fig. 21, 22). Notably, water loss and wounding induced only minor upward shifts in the impedance magnitude spectrum, with small and distinct variation trends. These findings further validate that the observed impedance changes in the transgenic A. thaliana are specifically induced by the immune response rather than other factors.”

3. Please test the e-tattoo's ability to detect NLR-mediated immune responses in plants infected with living virulent and avirulent pathogens. This would validate its practical application in studying immune responses to biotic stress and help determine the specificity

of EIS signals for NLR-mediated responses.

Reply:

We appreciate the reviewer's insightful comment. Plant immune responses to virulent and avirulent pathogens are highly complex, involving both pattern-triggered immunity (PTI) and effector-triggered immunity (ETI). These responses rely on multiple immune receptors, including plasma membrane-localized PRRs and intracellular NLRs, making them far more intricate than purely NLR-mediated immunity. The functional overlap between these receptor types further complicates the interpretation of impedance changes, as it becomes challenging to attribute specific signals solely to NLR activation. To address this, we have dedicated substantial effort to establishing the *Dangerous Mix (DM)* autoimmunity system, which induces a purely NLR-mediated immune response without requiring pathogen treatment [R1]. This system provides a well-defined and controllable platform for specifically investigating NLR-mediated immunity, eliminating confounding factors associated with other immune pathways.

To address the reviewer's concern regarding the specificity of the EIS signal, we have included new Supplementary Figures 37-39 and Supplementary Note 2, which demonstrate the feasibility of the e-tattoo in tracking distinct physiological changes associated with two representative NLR-mediated immune responses.

To ensure precise control over immune activation, we established stable *A. thaliana* transgenic lines that allow genetic induction of NLR-mediated responses with minimal external intervention. Using an ethanol-inducible promoter (*pAlcA*), we selectively activated one incompatible, namely *DANGEROUS MIX (DM)* partner, enabling precise temporal control over NLR-triggered immunity. Specifically, we generated two ethanol-inducible lines:

- **DM6-DM7 case:** Stable transgenic lines carrying *pAlcA::RPW8.I^{KZ10}* in Mrk-0 accession background, which activates CNL-mediated immune responses
- **DM10-DM11 case:** Stable transgenic lines carrying *pAlcA::DM10^{TueScha-9}* in Cdm-0 accession background, which activates TNL-mediated immune response

Due to the response kinetics, our study primarily focuses on CNL (coiled-coil NLR)-mediated immune responses, specifically the DM6-DM7 case. We have provided substantial evidence demonstrating the feasibility of using the e-tattoo to monitor this type of NLR-mediated response (Fig. 5, 6, Supplementary Fig. 23-36). For this revision, we have included additional data showing that the e-tattoo successfully detects impedance changes associated with TNL (Toll/interleukin-1 receptor-like NLR)-mediated immune activation in the Cdm-0 T₃ [*pAlcA::DM10^{TueScha-9}*] line (DM10-DM11 case) [R2].

TNL- and CNL-mediated immune responses exhibit distinct signaling mechanisms and kinetics, resulting in different impedance signatures. CNL-mediated immune responses, such as those triggered by AvrRpt2 [R3], induce rapid hypersensitive response (HR) and cell death. Recent studies have demonstrated that ZAR1, a well-characterized CNL, forms a calcium-permeable resistosome, leading to a swift ion influx and rapid HR within hours post-activation [R4]. This provides mechanistic support for the rapid impedance decrease we observed.

In contrast, TNL-mediated responses, such as those triggered by AvrRps4, involve a slower yet prolonged activation phase, requiring downstream components like EDS1 and NRG1 [R3]. The extended kinetics of TNL responses align with our impedance measurements and reinforce the conceptual distinction between these two immune receptor classes. For CNL-mediated immune responses, we observed a sharp decrease in impedance magnitude within ~3 hours post-induction. In contrast, TNL activation displayed a different pattern: impedance magnitude gradually increases, reaching a peak 4-6 dpi, followed by a sharp decline, indicative of severe cell death. This prolonged response reflects the slower propagation of TNL-mediated immunity compared to CNL.

By capturing these distinct impedance dynamics, the e-tattoo provides a clear and specific readout of NLR-mediated immune responses. This validation strengthens the specificity of leaf EIS signals in detecting NLR activation. Compared to traditional pathogen-based approaches, our genetically controlled model offers a more precise and reproducible method for studying NLR-mediated immunity, ensuring that observed impedance changes can be attributed directly to NLR activation. Thus, while testing the e-tattoo in pathogen-infected plants remains an important future direction, our current study establishes a rigorous and controlled framework for characterizing NLR-specific impedance signatures, providing a solid foundation for future investigations into biotic stress responses.

In response to the reviewer's concern, we have expanded the Discussion section to highlight this critical future research avenue, emphasizing how integrating pathogen-based studies with our approach could further enhance the applicability and specificity of EIS for monitoring plant immune responses.

"While our current study provides a rigorous and controlled framework for characterizing NLR-specific impedance signatures, this work serves as a proof of concept, laying the foundation for future studies applicable to a broader range of immune responses and stressors. An important future direction involves investigating the e-tattoo's performance in plants infected with virulent and avirulent pathogens. Such studies would further validate its practical application in monitoring biotic stress responses and broaden the scope of its use in plant immunity research."

[R1] Chae, Eunyong, et al. "Species-wide genetic incompatibility analysis identifies immune genes as hot spots of deleterious epistasis." *Cell* 159.6 (2014): 1341-1351.

[R2] Barragan, Ana Cristina, et al. "A truncated singleton NLR causes hybrid necrosis in *Arabidopsis thaliana*." *Molecular biology and evolution* 38.2 (2021): 557-574.

[R3] Saile, Svenja C., et al. "Two unequally redundant "helper" immune receptor families mediate *Arabidopsis thaliana* intracellular "sensor" immune receptor functions." *PLoS biology* 18.9 (2020): e3000783.

[R4] Bi, Guozhi, et al. "The ZAR1 resistosome is a calcium-permeable channel triggering plant immune signaling." *Cell* 184.13 (2021): 3528-3541.

4. *I would also suggesting testing eTattoo on other plants (closer to crop species), such as *Nicotiana benthamiana* and tomato with virulance/avirulance pathogens.*

Reply:

We thank the reviewer for their constructive suggestion. We fully agree that demonstrating the versatility of the proposed e-tattoo and its capability for EIS monitoring across diverse plant species is essential to establishing its broader applicability. In our study, we have already tested the e-tattoo on a wide range of plant species (Fig. 4a–i), including *A. thaliana*, *Nicotiana benthamiana*, *Epipremnum aureum* (pothos), *Coleus scutellarioides*, and *Hypoestes phyllostachya*. We have added a new Supplementary Fig. 12 to demonstrate the applicability of the e-tattoo on crop species, such as *Brassica rapa* (oilseed), and *Ipomoea batatas* (sweet potato). These species were carefully selected to represent diversity in taxonomic classification and leaf morphology, showcasing the e-tattoo's exceptional adaptability for EIS monitoring across various plant types.

Supplementary Fig. 12. EIS data collected on leaves from crop species. a-b, Bode plots acquired with e-tattoo on the leaf of sweet potato and *Brassica rapa*, respectively.

We would also like to emphasize that the primary novelty of our work lies in the device design, developing an ultrathin, substrate-free, and highly conductive electronic tattoo (e-tattoo) specifically engineered for plants. The e-tattoo offers unique features such as excellent conformality, conductivity, optical transparency, and biocompatibility, enabling long-term, non-invasive EIS monitoring without damaging delicate plant tissues. This device detects immune responses through EIS, which is widely recognized as a powerful tool for analyzing the molecular composition and physical structure of tissues. However, the application of EIS for non-invasive monitoring of plant immune responses has been hindered by the incompatibility of conventional electrodes with fragile plant tissues. Our work addresses this gap by introducing a plant-specific e-tattoo with favorable properties, enabling the successful monitoring of plant immune responses using EIS.

As a proof-of-concept, we investigated NLR-mediated immune responses in *A. thaliana*. We believe this foundational demonstration paves the way for future applications of this technology to monitor immune responses in other plant species. Given EIS's well-established capabilities in assessing molecular composition and tissue structure, and our demonstration of the e-tattoo's applicability across diverse plant species for EIS, the broader applications of this technology in the future is straightforward.

While our current work emphasizes the development and validation of this technology, to further address the reviewer's concern, we have also explored another type of NLR-mediated immune response, i.e., TNL-mediated immune responses. Our results successfully capture distinct differences in the development speed and intensity of TNL responses compared to CNL responses, underscoring the versatility and sensitivity of the e-tattoo. These results have been included in the new Supplementary Fig. 37-39 and Note 2.

Additionally, we have expanded the Discussion section with the potential application and the prospects of the e-tattoo for broader use in agriculture, particularly through its integration with machine learning. This addition provides a forward-looking perspective, highlighting the transformative impact of this technology on plant health monitoring.

“To enhance the specificity of this technique, machine learning could play a pivotal role in differentiating multiple stressors or immune responses by analyzing complex EIS patterns. Advanced algorithms could extract nuanced features from impedance data, enabling more precise characterization of plant responses under varying conditions. This synergy between the plant e-tattoo and machine learning presents a promising pathway for developing a high-throughput, intelligent platform for studying plant immunity and stress physiology.”

Again, we thank all the reviewers for their constructive feedback. We believe that the manuscript is much improved as a result.

To the reviewers,

We thank all the reviewers for their insightful and constructive comments on our manuscript. Below, please find a point-to-point response to the comments. As detailed below, we have extensively revised the main text to address the concerns raised, which are highlighted in blue in the revised manuscript. We believe that the manuscript is much improved as a result.

Reviewer #1 (Remarks to the Author):

In this revised manuscript, the authors did a great job on responding to the previous questions. The comments are properly addressed by either providing more explanations (many of them have actually been incorporated into the main manuscript, rather than a simple reply in the response letter) and extensive new experiments to confirm the results. There are no further comments from this reviewer.

Reply:

We sincerely appreciate the reviewer's positive feedback and recognition of our efforts in addressing the previous concerns. Your insightful comments have greatly contributed to improving our manuscript. Thank you for your time and thoughtful review.

Reviewer #2 (Remarks to the Author):

Reply:

We thank the reviewer for accepting our responses and the contribution on reviewing the paper.

Reviewer #3 (Remarks to the Author):

Authors were able to address all my comments and questions.

Reply:

We sincerely appreciate the reviewer's thoughtful review and valuable comments, which have significantly enhanced our manuscript. Thank you for your time and effort.

Reviewer #4 (Remarks to the Author):

The manuscript has been extensively improved, and I appreciate the authors' efforts in addressing previous concerns. Before publication in Nature Communications, I recommend adding two additional datasets to further strengthen the study.

Reply:

We sincerely appreciate the reviewer's positive feedback and recognition of our efforts in addressing the previous concerns.

Firstly, the authors have clearly demonstrated that TNL- and CNL-mediated immune responses exhibit distinct impedance signatures. To complement this, please provide data showing when cell death is induced upon ethanol activation of DM10 in the Cdm-0 background.

Reply:

We sincerely appreciate the reviewer's insightful comment and the opportunity to clarify this point. To address this, we have provided experimental data demonstrating that ethanol-induced DM10^{TueScha-9} expression in the Cdm-0 background successfully triggers cell death (Fig. R1a). This is evident from visible symptoms observed in ethanol-treated plants compared to mock-treated controls at 4 days post-induction (dpi).

Additionally, we have included a conductivity assay (Fig. R1b) to quantify ion leakage over time, serving as an indicator of cell death progression. Leaf discs from ethanol-induced transgenic Cdm-0 plants exhibit a clear increase in conductivity compared to both mock-treated transgenic plants and ethanol-treated wild-type Cdm-0 plants, confirming that DM10^{TueScha-9} expression induces cell death in the Cdm-0 background. Notably, conductivity starts to rise at approximately 6 hours post-induction (hpi), indicating the onset of cell death, which occurs more gradually than in Mrk-0.

The investigation of Cdm-0 and DM10^{TueScha-9} expression is still ongoing and has not yet been published. Therefore, we have provided this experimental data in the response letter rather than in the supplementary files. We hope this additional dataset sufficiently demonstrates the cell death phenotype associated with DM10^{TueScha-9} expression in the Cdm-0 genetic background.

Fig. R1. Ethanol-induced DM10^{TueScha-9} expression triggers cell death in the Cdm-0 background.

(a) Representative images of 3-week-old Cdm-0 T₂ plants carrying the pAlcA::DM10^{TueScha-9} construct. Plants were either mock-treated with water (Mock) or treated with 1% ethanol to induce DM10^{TueScha-9} expression. Photos were taken at 4 days post-induction (dpi). Scale bar = 1 cm. (b) Ion leakage measurements over time for the Cdm-0 and transgenic DM10^{TueScha-9} plants.

Secondly, as the authors noted in their reply, plant pathogens trigger both PTI and ETI. It is important to assess whether e-tattoo technology can differentiate these two immune responses, demonstrating its specificity and potential application in plant immunity research. While the authors have tested the effect of wounding on impedance signatures, I recommend additional experiments using flg22 or other well-characterized PAMPs to determine whether this method can reliably distinguish PTI from ETI.

Reply:

We sincerely appreciate the reviewer’s insightful comment and suggestion for future research direction. We agree that both PTI and ETI play crucial roles in plant immunity and are important areas of study. In our work, we choose ETI as a case to demonstrate the capability of our sensing system for immune response monitoring due to two main reasons. Firstly, the ETI-associated cellular changes are rapid and strong, which could induce large impedance variations to be easily detectable. Secondly, our collaborators have established stable *A. thaliana* transgenic lines that allow genetic induction of NLR-mediated responses with minimal external intervention, which has provided precise temporal control over NLR-triggered immunity.

PTI, triggered by Pattern Recognition Receptors (PRRs) at the cell surface, generally exhibits a weaker, slower, and more transient response compared to ETI [R1-R3]. PTI is known to involve ROS production, cytosolic calcium elevation, callose deposition, and defense protein accumulation, all of which alter the electrical properties of leaf tissues, as highlighted in a recent study on flg22 sensing [R4]. Therefore, we hypothesize that these changes could be detectable through EIS analysis with the e-tattoo. However, the relatively weaker PTI response is likely to result in smaller bioimpedance variations, posing additional challenges for our current sensing system. Addressing this would require significant optimizations, including refining the electrode layout, measurement protocols, and PAMP treatment methods. This is not a straightforward trial-and-error experiment but rather a rigorous optimization process requiring substantial efforts from both engineering and plant science perspectives in the future.

While we recognize the importance of differentiating PTI from ETI as a future research direction, we would like to emphasize that the key novelty and contribution of our work lie in the device design, developing an ultrathin, substrate-free, and highly conductive electronic tattoo (e-tattoo) specifically engineered for plants. This e-tattoo offers unique advantages, including excellent conformability, high conductivity, optical transparency, and biocompatibility, enabling long-term, non-invasive EIS monitoring without damaging delicate plant tissues. Although EIS is widely recognized as a powerful tool for analyzing molecular composition and tissue structure, its application in non-invasive plant monitoring has been hindered by the incompatibility of conventional electrodes with fragile plant tissues. The advance of our work is to address this gap by developing a plant-specific e-tattoo with favorable properties, successfully enabling continuous EIS-based plant physiological status monitoring.

Therefore, while differentiating PTI from ETI is undoubtedly an insightful research direction, it is beyond the scope of this study. As a proof of concept, our e-tattoo has demonstrated the ability to monitor NLR-mediated immune response in *A. thaliana*, leveraging a well-established genetic model that ensures precise and reproducible analysis. As requested by the reviewer's previous comments, we have added discussions on wounding, dehydration, and another type of NLR-mediated immune response, all of which validate the robustness and potential of this technology for broad plant monitoring applications.

Nevertheless, we sincerely appreciate the reviewer's suggestion, and we have further expanded the Discussion section of our manuscript to acknowledge this promising avenue for future investigation:

"Beyond differentiating specific NLR-mediated responses, future work could focus on optimizing the e-tattoo's sensing capability to differentiate between pattern-triggered immunity (PTI) and effector-triggered immunity (ETI). PTI, which is initiated by pattern-recognition receptors (PRRs) at the cell surface, generally leads to more transient and moderate physiological changes than ETI. Given that PTI responses involve ROS bursts, cytosolic calcium elevation, and callose deposition, all of which can affect the electrical properties of plant tissues, we hypothesize that EIS monitoring with e-tattoos could potentially capture these

dynamic changes. However, detecting PTI effectively remains a challenge due to the relatively weaker and shorter-lived nature of PTI responses, which may lead to subtle impedance variations. Enhancing sensitivity will require optimization in several key areas, including Pathogen-associated molecular pattern (PAMP) treatment strategies, electrode design and placement, measurement protocols, and signal processing techniques. Addressing these challenges could improve the e-tattoo's ability to distinguish between PTI and ETI, further expanding its applicability in plant immunity research."

[R1] Jones, Jonathan DG, and Jeffery L. Dangl. "The plant immune system." *Nature* 444.7117 (2006): 323-329.

[R2] Ngou, Bruno Pok Man, et al. "Mutual potentiation of plant immunity by cell-surface and intracellular receptors." *Nature* 592.7852 (2021): 110-115.

[R3] Tsuda, Kenichi, and Fumiaki Katagiri. "Comparing signaling mechanisms engaged in pattern-triggered and effector-triggered immunity." *Current opinion in plant biology* 13.4 (2010): 459-465.

[R4] Furch, Alexandra CU, et al. "Transformation of flg22 perception into electrical signals decoded in vasculature leads to sieve tube blockage and pathogen resistance." *Science Advances* 11.9 (2025): eads6417.

Again, we thank all the reviewers for their constructive feedback. We believe that the manuscript is much improved as a result.

To the reviewers,

We thank all the reviewers for accepting our responses to their comments and their contribution on reviewing the paper.

Reviewer #4 (Remarks to the Author):

The authors have satisfactorily addressed all the points I raised, and the manuscript is now ready for publication in Nature Communications.